
# Monitoring surface water quality using social media in the context of citizen science

Hang Zheng[1], Hong Yang[1], Di Long[1], Hua Jing[1]

[1]State Key Lab of Hydroscience and Engineering, Dept. of Hydraulic Engineering, Tsinghua Univ., Beijing 100084, China

*Correspondence t8o*: Hang Zheng (zhenghang@tsinghua.edu.cn)

**Abstract.** Surface water quality monitoring (SWQM) provides essential information for water environmental protection. However, SWQM is costly and is limited in terms of equipment and sites. The global popularity of social media and intelligent mobile devices with GPS and photograph functions provide immense opportunity for citizens to monitor surface water quality. This study aims to establish and demonstrate a method to monitor surface water quality using social media

platforms. A WeChat-based application platform is built to collect water quality reports from volunteers. Results show that the monitoring reports are reliable if the volunteers are trained. The key application functions and the methods for data washing and volunteer recruitment are also discussed in this study. The proposed framework and method can provide a mechanism to collect water quality data from citizens and offer a primary foundation for big data analysis in future research.

**Keywords.** Water quality; monitoring; social media; volunteered geographic information.

**1 Introduction**

Surface freshwater is a finite resource that is essential for human and ecosystem existence. An adequate quantity and quality of water are essential for sustainable development (Khalil and Ouarda, 2009). However, many surface water systems have been contaminated by treated or untreated wastewater discharged from domestic, industrial, and agricultural water users. Water quality has become an important component of the global water scarcity and water crisis.

The degradation of the surface water system has increased the need to determine the status of water quality to detect water pollution, thereby providing scientific guidance for water resources management (Wang et al., 2014). Water quality monitoring refers to the acquisition of quantitative and representative information on the physical, chemical, and biological characteristics of water bodies over time and space (Sanders et al. 1983; Strobl and Robillard 2008). Monitoring sites, frequency, variables and instruments, and trained/educated field personnel are required in a water quality monitoring

network. Evidently, establishing a surface water quality monitoring (SWQM) network in a broad area is costly (Horowitz, 2013). For example, the US Geological Survey runs the Mississippi River Basin monitoring network to address the gulf coast land loss and hypoxia. Currently, collecting a single sample at a site costs between US$4,000 and US$6,000. Analyses of various physical/chemical parameters add US$1,500 to US$2,000 more per sample (Horowitz, 2013). These costs limit the sample numbers/sites that can be monitored. The limited resources necessitate the monitoring program's installation of a





limited number of monitors on sites and samples at regular temporal intervals. These limitations reduce the capability of the monitoring program to disclose illegal pollution activities, such as hidden sewage dumping, which tend to occur in areas that are distant from the monitoring sites or at a time when no sampling has been conducted. For example, several Chinese industrial facilities dump sewage water discharge in rivers in midnight to avoid detection (Wei, 2013).

A participatory monitoring approach by citizens could fill the spatiotemporal gaps of the current monitoring network. Wei (2013) explained that two ordinary citizens recorded the sewage dumping in a river near their home in China from 2004 to 2007. Collecting samples and taking photos for four years enabled the two citizens to document dozens of instances of discoloured, foul-smelling, or warm water in the river. These voluntary records imply that citizens who are affected by the sewage discharge have a strong motivation to monitor and report the pollution. If a sufficient number of volunteer reporters

(e.g., the two citizens) come forward, then the hidden sewage dumping and pollution can be detected with considerably low cost.

Water quality can be defined in terms of one variable to hundreds of compounds and for multiple usages (Khalil et al., 2010). To collect the evidence of hidden or midnight sewage dumping, what the citizens see and photograph appear to be effective. Citizens without professional equipment for water quality analysis can describe the physical characteristics of the water (e.g.,

colour, smell, and temperature) to assess the water quality and disclose the pollution. Voluntary reporting is more flexible and effective and less costly compared with the traditional monitoring program operated by the government.

Thus, we argue that the volunteered geographic information (VGI) in the citizen science context provides a proximate sensing solution for water conservation issues. VGI has recently provided an interesting alternative to traditional authoritative information from mapping agencies and corporations (Goodchild and Glennon, 2010). VGI is also identified as

"collaboratively contributed geographic information" (Bishr and Mantelas, 2008) in the context of participatory GIS (McCall, 2003), crowdsourcing GIS (Goodchild and Glennon, 2010), participatory planning (Seeger, 2008), and citizen science (Tulloch, 2008).

Citizen science, which is an indispensable means of combining ecological research with environmental education and natural history observation, ranges from community-based monitoring to the use of the Internet to "crowd source" various scientific

tasks (i.e., from data collection to discovery) (Dickinson et al., 2012). Citizen science is the process whereby citizens are involved in science as researchers (Linda E. Kruger, Margaret A. Shanno, 2000). A citizen scientist is a volunteer who collects or processes data as a component of a scientific enquiry. Currently, citizen scientists participate in projects on climate change, invasive species, conservation biology, ecological restoration, water quality monitoring, population ecology, and different types of monitoring (Silvertown, 2009). VGI in the context of citizen science can be produced immediately and

may determine environmental changes nearly rapidly as they occur. VGI is recognized as an innovative approach for an improved environmental governance by fostering accountability, transparency, legitimacy, and other dimensions of governance (McCall, 2003).

Several recent studies have provided the beginnings of the literature on this new approach. VGI has been applied to numerous research and business domains, particularly in detecting, reporting, and geo-tagging disasters, including



earthquakes (Kim, 2014), floods (Perez et al., 2015), hurricanes (Bunce et al. 2012; Virtual Social Media Working Group 2013), wildfires (Slavkovikj et al., 2014), tsunamis (Mersham, 2010), and storms (Lwin et al., 2015). VGI has successfully increased public knowledge on emergency situations and provided a novel and effective approach for disaster warning and management. (Sakaki, 2009)built an earthquake detection system in Japan by monitoring reports submitted by citizens

through tweets. This system promptly detects earthquakes and sends e-mails to registered users within a minute (occasionally within 20 seconds) after earthquakes are detected. Notifications are delivered considerably faster than the announcements that are broadcasted by the Japan Meteorological Agency (JMA). On average, a JMA announcement is broadcasted 6 minutes after an earthquake. (Tang et al., 2015) descriptively evaluated the strengths, weaknesses, opportunities, and threats of VGI in managing the California drought in 2014 and provided a relatively overall description of

VGI's role in disaster management. Apart from being a practical tool for events detection, VGI provides a new level of interaction, participation, and engagement to citizens for environmental governance (Werts et al., 2012). VGI also creates a new paradigm to investigate the self-aware, self-adapting, and self-organizing socio-technical system that combines people, mobile technology, and social media in a complex network of information (Perez et al., 2015).

One of the major obstacles for using VGI is its unknown quality. The general population is untrained to make the specific

observations necessary in environmental management, and may intentionally or unintentionally supply erroneous information. Data quality is often unknown, and data sampling is frequently dispersed and unstructured. Similar concerns are often expressed regarding many other types of information provided by amateurs, thereby reflecting the profound association that society makes between qualifications, institutions, and trust (Goodchild and Glennon, 2010). Nevertheless, several grounds are used as basis for believing that the VGI quality can approach and even exceed that of authoritative sources

(Goodchild and Glennon, 2010). Fore et al. (2001) trained volunteers to collect benthic macro invertebrates using professional protocols, and detected no significant difference between the field samples collected by volunteers and those collected by professionals. Citizen volunteers who are properly trained can collect reliable data and make stream assessments that are comparable with those made by professionals. Data collected by volunteers can supplement information used by government agencies to manage and protect rivers and streams (Fore et al., 2001).

Researchers have provided the literature on the theory and application of VGI, although studies on water pollution detection and water quality monitoring through VGI are limited. Training ordinary citizens on water sampling and quality analysis in the laboratory is costly. Meanwhile, citizens can observe and perceive polluted water and disclose water pollution activities. Volunteers can photograph the water, describe its physical characteristics, and share the information online. In this case, guidance is necessary to assist volunteers express what they observe in a logical, clear, and structured manner. A mechanism

that can motivate volunteers to continuously produce data is also critical for VGI-based water quality monitoring.

Social media, such as Twitter, Facebook, Sina Weibo, and Tencent Weixin (the popular Chinese version of Twitter), is capable of guiding and providing incentives to volunteers through real-time online communication among volunteers. In recent years, social media has become a major communication channel in our society (Jiang et al., 2015). Social media refers to Internet-based applications that enable people to conduct online communications intended for interaction, community



input, and collaboration (Lindsay, 2011). Social media enables information sharing from multiple parties on computers and mobile devices, particularly through social networking sites (e.g., Facebook, YouTube, and Twitter), texting, chat rooms, discussion forums, and blogs (Tang et al., 2015). Social media builds on the ideological and technological foundations of Web 2.0, and enables the creation and exchange of user-generated content (Kaplan and Haenlein, 2010). The major functions

of social media in the environmental management processes include one-way information sharing, two-way information sharing, situational awareness, rumor control, reconnection, and decision-making (Tang et al., 2015). Jiang et al. (2015) effectively monitored the dynamic changes of air quality in large cities by analyzing the spatiotemporal trends in geo-targeted social media messages with comprehensive big data filtering procedures. (Werts et al., 2012) developed an integrated framework for combining WebGIS technologies, data sources, and social media for future use in soil and water

conservation. Werts et al. (2012) established a website called AbandonedDevelopments.com to collect VGI and detect the sediment pollution of abandoned structures in upstate South Carolina.

The advertisement, instruction, and guidance for water quality monitoring can be spread extensively and delivered to the potential volunteers' mobile devices directly through social media platforms. The observed sewage dumping or water pollution activities can be diffused rapidly in social media networks and call the government's attention. Social media

provides the platform for volunteers to present, discuss, and communicate their criticism, anger, and solutions to the water pollution they observe. Communication and mutual encouragement provide a strong motivation for volunteers to monitor water quality and persistently share their observations. Furthermore, the discussion on the pollution activities reported in social media networks increases the opinion's pressure on the government to solve the problem. Government feedback can also be promptly disseminated to volunteers through social media. Timely dissemination of government feedbacks also

motivates volunteers to continuously monitor water quality. Today, volunteers are equipped with digital cameras, GPS, digital maps, and other resources. Multiplying the resources of the average empowered citizen by the population of the city results in an astounding ability to create and share information (Goodchild and Glennon, 2010).

This study aimed to establish an approach to monitor surface water quality through volunteered citizen scientists. An App platform basing on social media was built to collect the water quality information. Data were cleansed, quantified, and

compared with the official monitoring data from the local water authority. Results revealed a spatiotemporal relationship between social media messages and real-world environmental status changes. The results also suggested new methods to monitor water pollution using VGI and social media in the future. This paper is organized as follow. Methodology is presented in Section 2, followed by a case area in Section 3. All monitoring reports obtained across China are displayed in Section 4. The data quality and motivation of volunteers are discussed in detail in Section 5. Conclusions drawn from this

study are given in Section 6.



## 2 Methodology

A methodological framework was established to: (1) Collect sensory data of surface water quality from the volunteer citizens, who describe and photograph the water that they pass by or are close to. Descriptions and photos are sent to a data centre from the mobile devices of volunteers through a specific application in the social media platform. (2) Detect the

illegal/hidden sewage dumping that the official monitoring network hardly covers. Once sewage dumping reports from the volunteers are submitted, the data centre will transfer them to the mass media and water authorities after the credibility check.

### 2.1 Data type

Four indicators were adopted to describe the physical characteristics of water quality (see Figure 1). A total of 11 water color options were provided for volunteers to choose from, including red, orange, yellow, green, cyan, blue, purple, milky, pink,

black, and crystal. The second indicator is smell, which is quantified by the ranking scores made by volunteers. The volunteer is supposed to rank the smell of the sample from 0 to 10, with 0 implying a lack of odour and 10 implying a foul-smelling sample. The third indicator is turbidity in which a score of 10 means that the water is non-transparent and a score of 0 represents transparency. The fourth indicator represents the floating objects or floats on the water. If the water is completely covered by oil, forth, plastics, and rubbish, among others, the floats score is 0. By contrast, a score of 10 means

no floating object is present. The last item is an integrated assessment on the water quality, ranking from worst, very bad, bad, good, to excellent. Volunteers evaluate the water quality based on their perception.

### 2.2 App in social media platform

The Tsinghua Environment Monitoring Platform (TEMP, http://www.thuhjjc.com/) application was built based on WeChat public accounts. WeChat is a mobile text and voice messaging communication service developed by Tencent in China, and

was first released in January 2011. WeChat is one of the largest messaging applications in China. As of May 2016, WeChat has over a billion existing accounts and 700 million active users (Intelligence, 2016). WeChat provides text messaging, hold-to-talk voice messaging, broadcast messaging, video conferencing, photo and video sharing, and location sharing (Tencent, 2016). Moreover, WeChat enables users to register a public account, thereby enabling them to push feeds to subscribers, interact with subscribers, and provide them with services. Public accounts can be used as a service platform, such as hospital

pre-registrations, visa renewal, or credit card service (Wikipedia, 2016). WeChat provides functions for users to post images and texts, share music and articles, as well as comment and "like" in the Moments. Only the users' friends will be able to view the contents and comments of their Moments. In addition, WeChat supports payment and money transfer, thereby enabling users to use peer-to-peer transfer and electronic bill payment (Tencent, 2016).

Volunteer reporters should install WeChat in their mobile devices and log in the TEMP through their WeChat accounts.

Through TEMP, volunteers are able to summit reports together with the GPS position where the reported water is located. The location can either be automatically extracted from the devices or be input manually by the reporters. Volunteers can



tweet the reports to their friends, and post and comment on them in the Moments. TEMP also provides a function for ranking volunteers according to their contributions. Awards, such as cash through WeChat Payment, will be offered to top-ranking reporters. A computer-based website is also provided for the public to view and download the reports (See TEMP, http://www.thuhjjc.com.)

**2.3 Volunteer recruitment**

Two modes were used for volunteer recruitment. In Mode 1, TEMP was popularized from a central group to the general public (see Fig. 1). The university students recruited for the present study post the address of the TEMP two-dimension code in Moments and Chat Groups after they log in the TEMP through their WeChat accounts. Their friends who are interested in SWQM will click the address or scan the two-dimension code, and then be directed to TEMP. If these people log in and submit reports, then they become volunteers. Thereafter, they may share their reports and attract numerous volunteers to be involved. TEMP is not able to control when and where the monitoring reports come from. Data was scattered in Mode 1.

In Mode 2, a group of professional citizens were recruited to monitor the water quality in targeted sites. Professionals who work for environmental authorities and organizations were interviewed and convinced to register in TEMP. They were required to monitor water bodies that they are familiar with and regularly submit the reports through TEMP. In Mode 2, volunteers were recruited and data were instantly collected before an extensive range of citizens who were involved through Mode 1.

**2.4 Data analysis**

A method was established to quantitatively analyse monitoring reports. Smell, turbidity, floats, and integrated assessment reported by the volunteers were quantified and normalized between 0.0 and 1.0 according to their ranking scores. Water colour was used for data cleansing and rumour control through cross validation between the descriptions and photos. Table 1 shows the indicators and their value ranges.

**Table 1: Normalization of indicators**

| Report items | Indicator | Data type | Qualifications |
| --- | --- | --- | --- |
| Water color | $C$ | Text | None |
| Smell | $S$ | Score from 0 to 10 | 1.0–0.0 |
| Turbidity | $T$ | Score from 0 to 10 | 1.0–0.0 |
| Floats | $F$ | Score from 0 to 10 | 1.0–0.0 |
| Integrated assessment | $I_a$ | Grand from 1 to 6 | 1.0, 0.75, 0.5, 0.25, 0.0 |

The value of smell, turbidity, and floats ranges from 1.0 to 0.0. The integrated assessment's indicator is normalized across 1.0, 0.75, 0.5, 0.25, and 0.0, which correspond to the five grades of water quality assessment (i.e., excellent, good, bad, very bad, and worst).



$$Q = (P_a + I_a)/2 = [(S + T + F)/3 + I_a]/2, \tag{1}$$

where $Q$ represents the report's water quality assessment result. A preliminary assessment ($P_a$) is calculated through the average of $S$, $T$, and $F$. However, volunteers have already provided an integrated assessment ($I_a$) in the reports based on their perception of the water. The water sample's smell, turbidity, and floats were also used in the calculation to enhance the assessment's credibility.

The gauged water quality data from water authorities were also normalized for comparison. According to the Chinese standard of surface water quality issued by the Ministry of Environment (GB 3838-2002) in 2002, surface water quality is classified into five grades (Table 2). Three key indicators were adapted to normalize the gauged water quality, namely, the ingredient content of permanganate, $NH_3$-N, and dissolved oxygen. The ingredient content is transferred to the normalized value based on the water quality grades. Equation (2) shows this method.

**Table 2: Normalization of the gauged water quality**

| Water quality grade | I | II | III | IV | V |
|---|---|---|---|---|---|
| Index of Permanganate (mg L$^{-1}$) | <=2 | 2–4 | 4–6 | 6–10 | 10–15 |
| Qualification of Permanganate ($E$) | 1.00 | 0.75 | 0.50 | 0.25 | 0.00 |
| $NH_3$-N (mg L$^{-1}$) | <=0.15 | 0.15–0.50 | 0.50–1.00 | 1.00–1.50 | 1.50–2.00 |
| Qualification of $NH_3$-N ($N$) | 1.00 | 0.75 | 0.50 | 0.25 | 0.00 |
| Dissolved Oxygen (DO) (mg L$^{-1}$) | >= 7.5 | 7.5–6.0 | 6.0–5.0 | 5.0–3.0 | 3.0–2.0 |
| Qualification of DO ($D$) | 1.00 | 0.75 | 0.50 | 0.25 | 0.00 |

$$P = (E + N + D)/3, \tag{2}$$

where $P$ is the normalized value of the gauged water quality. Moreover, $E$, $N$, and $D$ are the normalized values of the indicators, namely, permanganate, $NH_3$-N, and dissolved oxygen, respectively. If the index of permanganate is below 2 mg L$^{-1}$, then the normalized indicator of $E$ is equal to 1.00. If the index of permanganate is between 2 and 4, then the value of $E$ is equal to 0.75. $Q$ and $P$ in the same day and on the same site are compared to validate the credibility of $Q$.





**Figure 1: Framework of monitoring water quality by VGI and social media.**



## 3 Case area

The Yellow River is the second largest river in China in terms of length and basin area, and is the seventh largest in terms of discharge (Chen et al., 2003). Originating in the Bayan Har Mountains in Qinghai Province of western China, the Yellow River flows through nine provinces and empties into the Bohai Sea. The Yellow River's total basin area is 742,443 km$^2$ (Bai et al., 2016). Sewage discharge in the Yellow River basin has heavily polluted water resources and damaged the function of water bodies and the ecological environment because of the continuous and rapid development of the social economy and increase in water resource exploitation. Among the 83 cross-sections surveyed during the Yellow River quality assessment in 2004, 72.3% felt into the Grade III quality standard. Meanwhile, among the water sources of cities along the Yellow River's mainstream, 70.0% still fall short of the standards of ground water source sites of drinking water, thereby posing a considerable threat to drinking water safety along the Yellow River (Yellow River Conservancy Commission, 2010).

The gauged water quality data at the Huayuankou hydrological station were used in the present study. The Huayuankou Station, with a catchment area of 730,036 km$^2$, was one of the key stations on the main reach and is located where the middle reach and lower reach are divided. The runoff at this point often reaches its maximum value because of the limited water flowing into the river channel downstream from this point given that the riverbed is higher than the land outside the banks (Fu et al., 2004). The hydrological regime at this station represents an overview of the hydrological regime of the entire river basin.

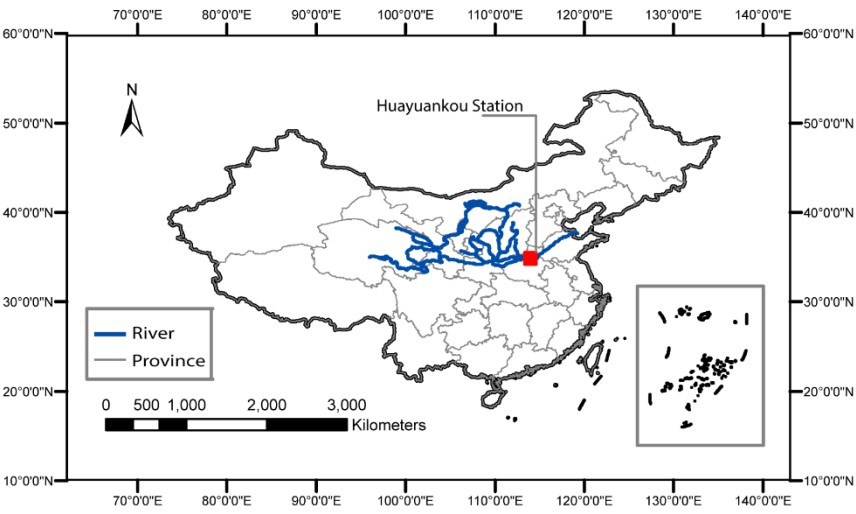

**Figure 2: Location of the Huayuankou Station and map of the Yellow River.**

Water quality reports at the Huayuankou Station were collected through Mode 2 (see Fig. 1). A group of professionals who are working in the Yellow River Conservancy Commission (YRCC) was recruited as volunteers. The group was supposed to visit the sites, describe the water quality, and regularly submit the reports through TEMP from 12 March 2016 to 13 April 2016. The data in the reports were normalized and compared with the gauged water quality data from the YRCC.



## 4 Results

TEMP received a total of 265 reports from volunteers in China between 12 October 2015 and 30 May 2016. Out of the 265 reports, 172 were validated and analyzed in the present study. Out of the 172 reports, 30 were about the water quality at the Huayuankou Station in the Yellow River.

### 4.1 Results at the Huayuankou Station, Yellow River, China

Figure 2 shows the water quality monitoring results at the Huayuankou Station. A total of 15 days were utilized, each of which involved monitoring data from volunteers and official gauge between 12 March 2016 and 13 April 2016. A strong positive correlation was determined between $Q$ and $P$. $Q$ is the normalized indicator, which was calculated based on the TEMP assessment reports. The reports related to the Huayuankou Station were generated by the YRCC professional staff members. The staff members were familiar with the water quality status at the Huayuankou Station, and provided substantially accurate data. The results imply that the water quality assessment through the citizens' sensory is compatible with the reality. The citizens' assessment is effective in representing the water quality status if the reporter/citizen is relatively trained or has the professional knowledge on water quality management. Similar evidence was also obtained by Fore et al. (2001), Monk et al. (2008), Flanagin and Metzger (2008), and Koss et al. (2005).

Figure 3 shows that the value of $Q$ is generally higher than that of $P$. This difference may be attributed to that citizens tend to overrate the water quality. People may recognize the water as good quality if no extreme smell or visible water pollutants were present. In addition, citizens observe and smell the water on the bank of the Yellow River when they submit the reports rather than through close observation on the water sample in the bottle. The distance between the reporters and water may affect the observers' senses.

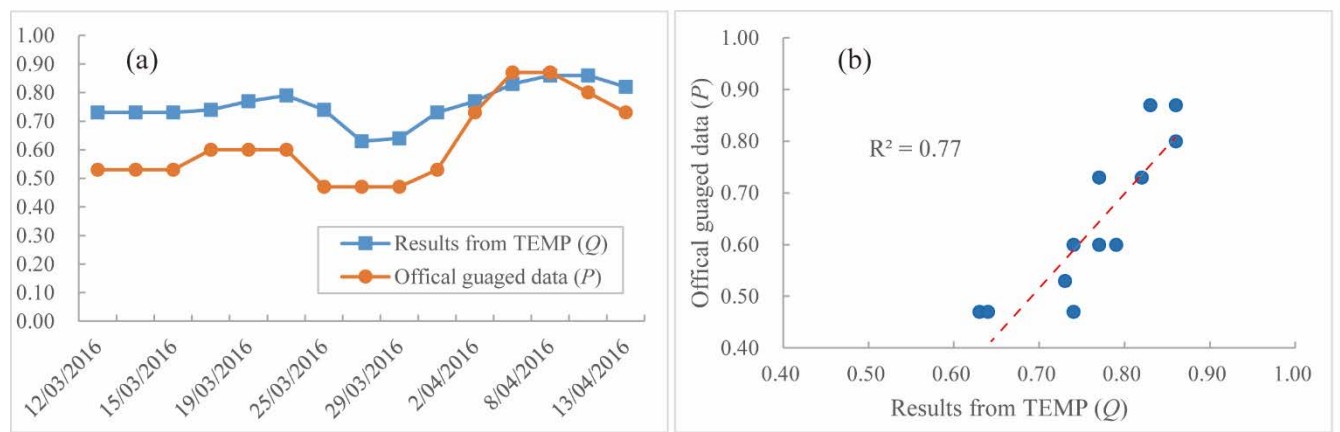

**Figure 3: Comparison of water quality assessment from citizens and authorities.**

Figure 4 shows the relationships between the official gauged data ($P$) and the smell ($S$), turbidity ($T$), and floats ($F$). The turbidity and floats of the water are considerably correlative with $P$ than the smell. The reason is that the turbidity and floats



of the water are considerably easy to observe through the reporters' eyes. The water's smell has limited contributions in monitoring because of the difficulty and subjectivity for citizens in distinguishing the water's smell at the river bank, particularly in a windy environment.

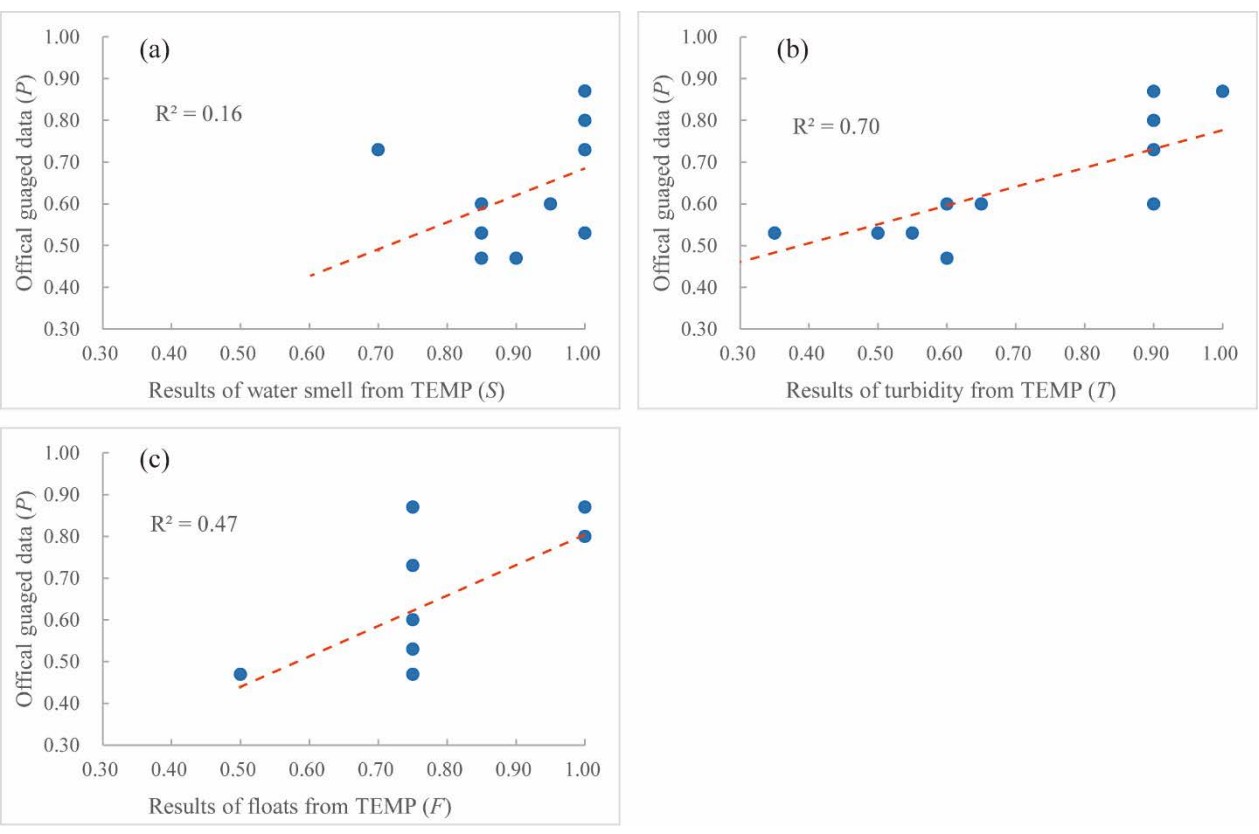

**Figure 4: Contributions of smell (Fig. 4(a)), turbidity (Fig. 4(b)), and floats (Fig. 4(c)) on water quality assessment from citizens.**

**4.2 Results across 29 provinces and cities in China**

Table 3 shows 172 validated reports across 29 provinces and municipalities in China. Henan Province and Beijing municipality have over 30 reports have over 30 reports. The reports from Henan Province were contributed by Henan-based YRCC professional volunteers. The reports from Beijing were mainly from the students of Tsinghua University, where the
10 research group of the present study is located. A total of 10 provinces have under 10 reports. A total of 13 provinces have only 1 report during the period. Meanwhile, no reports were provided from Tibet, Xinjiang, Hainan, Taiwan, Macau, and the South China Sea.

Table 3 shows that 82 out of 172 reports were submitted without the WeChat users' names indicated. TEMP users were allowed to submit their reports anonymously. A few volunteers were concerned about their privacy; thus, they submitted
reports anonymously. The number of anonymous reports implies the necessity of setting up an anonymous function in TEMP because volunteers care about their privacy when they disclose the water pollution activities around them. A total of 83



reports out of 172 had photos of water. Photos significantly increased the credibility of the reports by providing substantial information for water quality analysis. However, 50% of the reports had no photos. The volunteers were also concerned about the charge of the Internet data flow, through which they upload the photos without Wi-Fi. Additional incentive measures are required to encourage volunteers to upload the photos of water. Figure 5 and Figure 6 show the number of reports and anonymous reports and photograph reports, respectively.

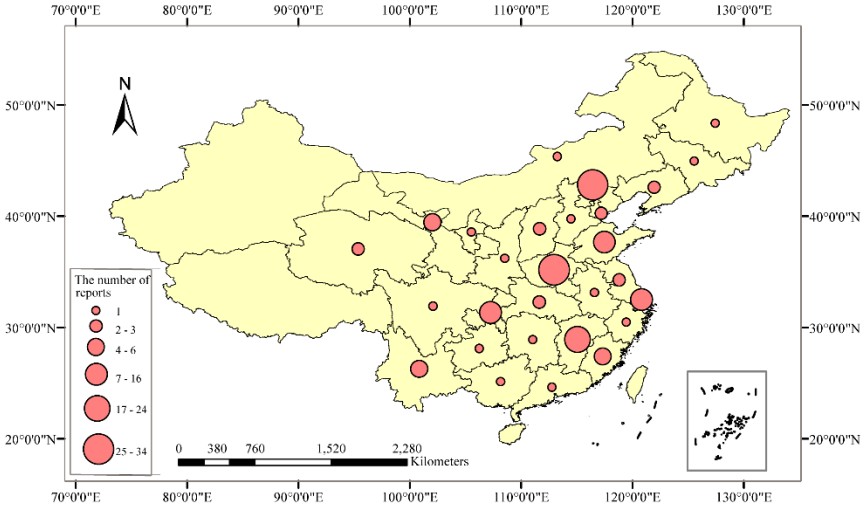

**Figure 5. Distribution of the reports across the provinces and cities in China.**

Note: No reports were provided from Tibet, Xinjiang, Hainan, Macau, Taiwan, and South China Sea.

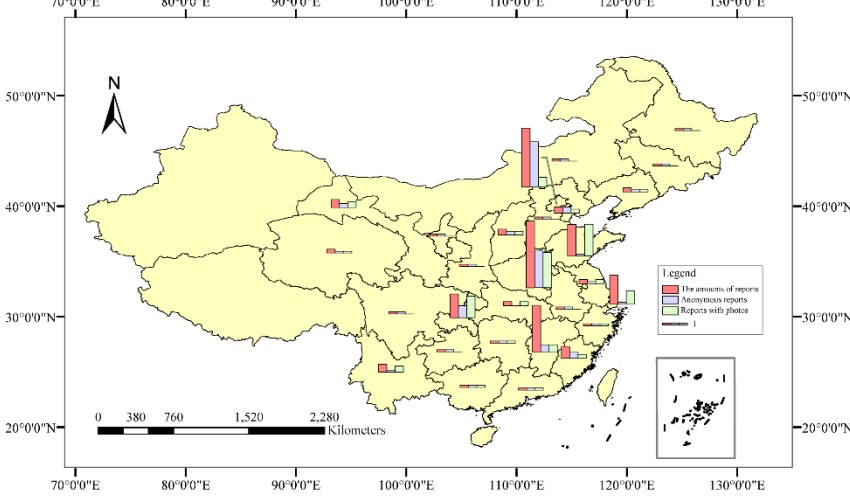

**Figure 6. Distribution of anonymous reports and photograph reports.**

Note: No reports were provided from Tibet, Xinjiang, Hainan, Macau, Taiwan, and South China Sea.





Table 3 shows the water quality assessment results. The assessment result of each province was calculated based on the volunteered reports from that province. The assessment is, however, unable to represent the overall situation of the surface water quality in a region because the coverage and frequency of the reports is insufficient at the current stage. Meanwhile, TEMP provides a practical tool for citizens to monitor the rivers and the lakes around them. An increasing number of volunteers will be involved in the future. The surface water quality of a region or a river basin can be depicted once enough volunteers and sufficient reports and photos are provided.

**Table 3: Number of reports across China**

| No. | Provinces | Number of reports | | | Assessment | No. | Provinces | Number of reports | | | Assessment |
|---|---|---|---|---|---|---|---|---|---|---|---|
| | | Total | Anonymous | Photographed | | | | Total | Anonymous | Photographed | |
| 1 | Henan | 34 | 19 | 18 | 0.74 | 18 | Guangdong | 1 | 1 | 1 | 0.75 |
| 2 | Beijing | 29 | 23 | 4 | 0.59 | 19 | Guangxi | 1 | 1 | 1 | 0.77 |
| 3 | Jiangxi | 24 | 4 | 4 | 0.69 | 20 | Guizhou | 1 | 1 | 0 | 0.77 |
| 4 | Shandong | 16 | 1 | 16 | 0.61 | 21 | Hebei | 1 | 1 | 0 | 0.73 |
| 5 | Shanghai | 15 | 1 | 7 | 0.74 | 22 | Hunan | 1 | 1 | 1 | 0.58 |
| 6 | Chongqing | 12 | 6 | 11 | 0.57 | 23 | Jilin | 1 | 1 | 0 | 0.78 |
| 7 | Fujian | 6 | 3 | 2 | 0.88 | 24 | Neimeng | 1 | 1 | 0 | 0.33 |
| 8 | Gansu | 4 | 2 | 3 | 0.75 | 25 | Ningxia | 1 | 1 | 0 | 0.80 |
| 9 | Yunnan | 4 | 1 | 3 | 0.90 | 26 | Shaanxi | 1 | 1 | 0 | 0.75 |
| 10 | Shanxi | 3 | 2 | 2 | 0.74 | 27 | Sichuan | 1 | 1 | 0 | 0.78 |
| 11 | Tianjin | 3 | 3 | 2 | 0.80 | 28 | Zhejiang | 1 | 1 | 1 | 0.58 |
| 12 | Hubei | 2 | 0 | 2 | 0.64 | 29 | Hongkong | 1 | 1 | 1 | 1.00 |
| 13 | Jiangsu | 2 | 1 | 2 | 0.77 | 30 | Xinjiang | / | / | / | / |
| 14 | Liaoning | 2 | 1 | 1 | 0.70 | 31 | Tibet | / | / | / | / |
| 15 | Qinghai | 2 | 1 | 1 | 0.88 | 32 | Hainan | / | / | / | / |
| 16 | Anhui | 1 | 1 | 0 | 0.80 | 33 | Taiwan | / | / | / | / |
| 17 | Heilongjiang | 1 | 1 | 0 | 0.55 | 34 | Macau | / | / | / | / |
| Total | | | | | | | | 172 | 82 | 83 | |

Table 4 shows three photograph reports. The algal bloom and water surface foam are shown in the photos. In Report 1, the volunteer submitted a photo of the river in Tsinghua University, Beijing, China. The river is polluted by the domestic sewage and suffers from eutrophication. Report 2 represents the water quality in an unidentified river located in Fei Town, Linyi City, Shandong Province. Photo 2 shows that the water surface is covered by algae and rubbish. The reporter assessed the water quality as very bad. Report 3 is from a coastal area, namely, Tianjing City. The reporter identified the color of the water as black and assessed its quality as bad. White foam was observed on the water surface, thereby implying that the quality of water is not good.





Among the 83 photos collected by TEMP during the period, the volunteers failed to monitor the scene, wherein the dumped sewage water was flowing to the river or lake. Sewage water dumping in China generally occurs at night and in a hidden place, which is hardly found by the volunteers. If there are sufficient volunteers aiming to disclose the hidden sewage dumping in a region, then the pollution activities can be determined by TEMP based on its current functions.



**Table 4: Examples of reports for pollution disclosure**

| No. | 1 | | | | | 2 | | | | | 3 | | | | |
|---|---|---|---|---|---|---|---|---|---|---|---|---|---|---|---|
| **Report** | Color | Smell | Turbidity | Floats | Assess. | Color | Smell | Turbidity | Floats | Assess. | Color | Smell | Turbidity | Floats | Assess. |
| | Green | 3.0 | 5.0 | 7.0 | VB | Green | 3.0 | 6.0 | 5.0 | VB | Black | 0.0 | 7.0 | 7.0 | B |
| **Photo** | | | | | | | | | | | | | | | |
| **Date** | April 10, 2016 | | | | | April 15, 2016 | | | | | May 1, 2016 | | | | |
| **Location** | Tsinghua University, Beijing City, China | | | | | Fei Town, Linyi City, Shandong Province | | | | | New coastal area, Tianjin City, China | | | | |

Note: B means bad; VB means very bad.



## 5 Discussion

The present study aimed to develop a method to monitor surface water quality through volunteer citizens using a social media application. A framework was established to guide the application design, volunteer recruitment, data collection, and report analysis. The TEMP application was built based on the social media platform called WeChat. Using the application,

TEMP users can describe and photograph the water in rivers and lakes following the TEMP instructions. Moreover, users can disclose the surface water pollution activities that affect their living and health.

A total of 172 validated reports were analyzed in this study. These reports are from 121 volunteers across 29 provinces and cities in China. The water quality on sites was assessed by the volunteers using their sense organs, particularly through their observations on water smell, turbidity, and floats on the water. People's sense on water quality varies, and different people

may provide different assessment reports on water from the same site. Comparing the assessment results across different sites is difficult because the citizens' reports are subjective somehow. Meanwhile, this situation will change if numerous volunteers and extensive reports are obtained. Data can be generated through satellite and aerial remote sensing, sensor systems streaming, informal online platforms, and other popular social media, such as Facebook and Twitter. Thereafter, the big data method (Hampton et al., 2013) can be applied to improve the accuracy of water quality monitoring if high data

density is present. Citizen scientists contribute to environmental or ecological data archive expansion by participating in collaborative projects (e.g., eBird or Wikipedia) (Kelling et al., 2009). These efforts are dedicated to addressing critical environmental challenges across spatiotemporal scales (Cha and Stow, 2015). The present study provides an approach for collecting citizen reports on water quality, which is the first step in applying the big data method in environmental governance (Perez et al. 2015; McCall 2003).

The credibility of the reports is the major concern of the present study. Regardless of the water quality assessment by citizens, identifying whether the reports are real and whether the volunteers generated the reports based on their observations is necessary rather than on false statements or rumours. Rumour control is significant when a water pollution activity is detected and reported in social media (Tang et al., 2015). The current study applied three criteria to validate the reports, including: (1) the report with the exact GPS location information of the site is regarded as credible. The sites' GPS

information is automatically abstracted from the reporters' mobile devices when they submit the reports. (2) If reports are submitted several times during a short period, and most of these reports are from the same volunteer on the same site, then the reports are deemed to have low credibility. These reports may be test reports from new volunteers. (3) Reports with photos are the most credible. A total of 265 reports are cleansed following these criteria, and 172 were validated in the present study. Further cross validation among different reports can be applied if huge amounts of data in a region are present.

Questionable reports can be cross-validated by the other volunteers' reports on the same site, and by time if the spatiotemporal density of the reports is sufficiently high.



The volunteers' motivation to consciously submit data is another major concern of the present study. Various VGI studies reveal that the volunteers' motivation is a key factor to the VGI program's success (Werts et al., 2012). Coleman et al. (2009), as cited in  (Werts et al., 2012, Page 817) revealed that the key motivators include altruism, professional or personal interest, intellectual stimulation, protection or enhancement of a personal investment, social reward, enhanced personal

reputation, outlet for creative and independent self- expression, and pride of location, among others. Several citizens may be motivated by the perceived instrumentality in promoting change (Hertel et al., 2003). Budhathoki et al. (2010), as cited in (Werts et al., 2012, Page 817), regarded fun, learning, and instrumentality as primary motivators for geographic information contributors, thereby noting that ''when contributors see their data appear visually on maps, they receive deep satisfaction.''

In the present study, the core members in the chat group who lead the communication and remind the volunteers are

beneficial in motivating the contributors. Moreover, certain economic incentives are quite effective in increasing the number of contributors and their contributions. Figure 7 shows the number of volunteers involved in the present study. The TEMP application was built and has been tested since April 2015. After the application development and testing, TEMP was promoted by the faculty, staff, and students at Tsinghua University though their WeChat Moments. The number of users steadily increased until the first official launch of TEMP in the ceremony conference for the establishment of the Tsinghua

Remote Sensing and Big Data Research Centre (http://hydrosky.org/Newscon/index/id/581/aid/544445113) in October 2015, when a sharp increase occurred.  Professionals and the public related to the research centre were observed to have registered in TEMP, thereby facilitating the increase in the number of users in October 2015. After the increase in the number of users, TEMP users continuously promoted and advertised the platform in their WeChat Moments, and attracted the involvement of additional users. The number of reports increased gradually because of the lack of incentives to the volunteers before March

20    2016.

In March 2016, Mode 2 was adapted to recruit YRCC volunteers and students in the universities in Beijing. Several volunteer chat groups were established in WeChat, and a core member was assigned in each group to lead the communication and remind users to submit reports. Consequently, the number of reports substantially increased after these activities. Since April 2016, economic incentives and rewards were adapted. The core members in the chat groups gave the

"Red Packet" money through WeChat Payment. The core members transferred the money in the chat groups, which was normally 100 RMB each time in a chat group. The "Red Packet" money can be obtained and shared by dozens of members, who point the "Red Packet" money picture on the screen before others do. The winner receives a random share of the total "Red Packet" money. The members who win the money in the chat groups were considerably motivated to submit reports, tended to invite their friends in, and encouraged them to submit data. After April 2016, these economic incentives

continuously motivated the volunteers and increased the number of reports.



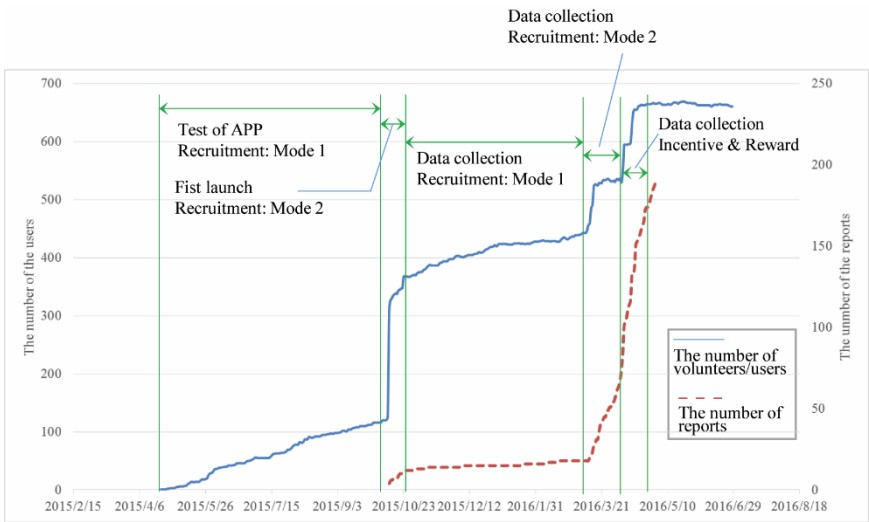

**Figure 7: Increase in the number of reports and reporters.**

The economic rewards were also provided in accordance with the volunteers' contributions, rather than their finger speed of pointing the "Red Packet" money picture on their mobile devices' screen. Figure 8 shows the distribution of the reports versus reporters. A total of 20 out of 39 reporters (excluding the anonymous reporters) submitted only 1 report each during the period. There were 3 reporters whose reports' number was larger than 10. A targeted reward by WeChat Payment was provided to the reporters with top-ranking submissions among all TEMP users since April 2016.

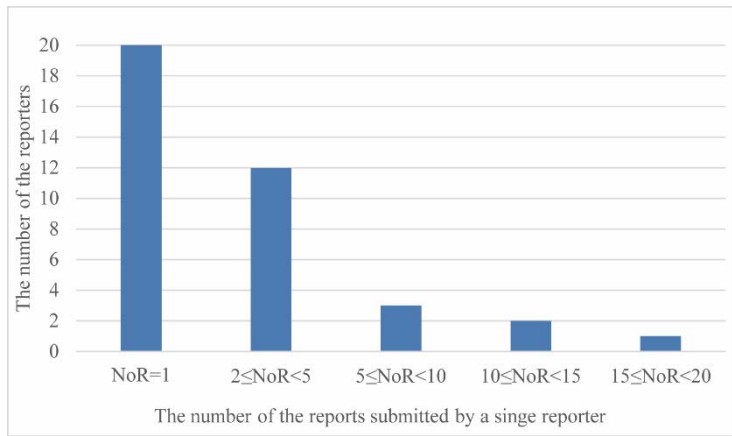

**Figure 8: Distribution of the reports versus reporters.**

Further research related to the present study may include efforts to collect additional data and recruit many active volunteers. Additional data sources, such as Twitter, Facebook, and Sina Weibo, could be involved (Jiang et al., 2015). People tend to post or tweet a text or a picture in social media to complain water pollution, and represent their lives to their friends, if they see a dirty river or lake (Kaplan and Haenlein, 2010). In this case, social media users do not intend to disclose a water pollution activity. However, users inadvertently provide the data for water quality monitoring. The text and photos about




water quality from Twitter, Facebook, and Sina Weibo can provide high density and massive data because of the presence of millions of active users in these social media platforms. This process saves the effort to recruit and motivate volunteers, although the data collected tend to be unstructured (Poser and Dransch, 2010).

TEMP facilitates the efficient method of contacting and guiding volunteers to submit the required structural report and data.

However, a considerable effort is necessary to recruit and motivate them. Economic incentives are demonstrated to be effective in the present study. Meanwhile, the use of incentives appears to be an unsustainable method. Cooperation with non-government organizations (NGOs) that are focused on environmental protection may provide another approach for volunteer recruitment. NGO members should be interested in TEMP and have an enduring and strong motivation to disclose water pollution activities.

The final solution for a durable motivation on SWQM is to establish a positive feedback between the citizens and the government (see Fig. 9). TEMP receives water pollution reports from volunteers. These reports are popularized in social media through communication among volunteers. After popularization through media, the pollution activities attract the attention of the mass media and are printed on newspapers or broadcasted on TV. The government is forced to reply and solve the problem because of public pressure. Once the disclosed pollution problems are solved and the water quality is

improved through government efforts, the volunteers will be extremely motivated to recommend TEMP to their friends and submit more data. Therefore, further research should include the cooperation with the mass media to realize the positive feedback ring shown in Fig. 9.

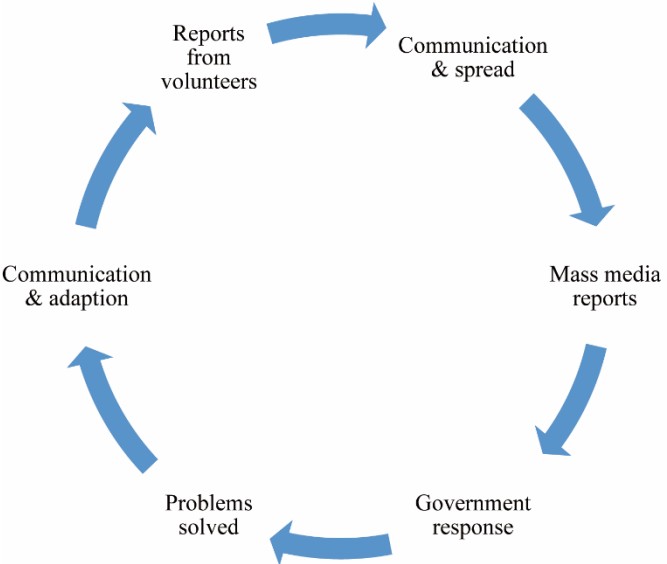

**Figure 9: Positive feedback ring for volunteer's motivation.**





## 6 Conclusion

A methodological framework to monitor surface water quality using social media is established in the present study, including the selection of the water quality indicators, application design guide, volunteer recruitment methods, data collection, cleansing, and analysis. The TEMP application was established based on a popular social media platform in China called WeChat. TEMP enables registered users to submit their descriptions and photos on rivers or lakes anonymously or non-anonymously with the automatically extracted GPS information of the sites from their mobile devices.

A total of 265 reports across 29 provinces and cities were received by TEMP between October 12, 2015 and May 30, 2016. Out of the 265 reports, 172 were validated and analyzed. The monitoring results on the Huayuankou Station in the Yellow River shows that the water quality monitoring results based on the reporters' vision and smell were relatively credible and comparable with the professionally gauged data from the water authority, if the volunteer reporters are trained or have certain professional knowledge on water quality analysis. The distribution analysis of reports across China indicates that the anonymous and photograph functions are quite essential for TEMP. Over 48% of the 172 reports are from anonymous users. Accordingly, people care about their privacy when they try to disclose a water pollution activity occurring within their vicinity. A total of 83 photos of rivers and lakes were collected through TEMP, and these photos provide extensive information for pollution detection.

Data quality and the users' motivation are also discussed. Three criteria were applied for data cleansing based on the location, time, and photos in the reports. Two modes for volunteer recruitment were used. Mode 2 is considerably suitable for increasing the number of volunteers within a short period. The economic incentive mechanism to targeted volunteers are also implemented in the present study. Economic incentives are effective in motivating the volunteers to contribute data under the guidance from the core members of the chat groups.

A substantially durable incentive mechanism beyond paying rewards to the volunteers is proposed at the end of the Discussion section. The positive feedback between the citizens, mass media, and government is argued to provide the solution for the volunteers' motivation in water quality monitoring.

Further research should include collecting more data and recruiting more motivated volunteers through implementing the positive feedback ring (see Fig. 9). The unpremeditated data referring to water quality from Twitter, Facebook, and Sina Weibo should be abstracted to increase the volume of data. The big data method is expected to be used with the availability of sufficient data.

## Data availability

The data underlying this research can be accessed publicly. All the data can be download from the website www.thuhjjc.com. This is a website established by the authors for displaying and downloading the data from TEMP platform.



**Author Contribution**

Dr. Hang Zheng designed the framework of this study, analysed the data and prepared the manuscript with contributions from all co-authors. Professor Hong Yang designed the interface of TEMP platform and contributed to the Discussion Section. Dr. Di Long carried out the data collection. Mr. Jing Hua developed main functions of the TEMP platform through computer programming.

**Competing interests**

The authors declare that they have no conflict of interest.

**Acknowledge**

This research is supported by National Natural Science Foundation of China (51479089) and (51323014); 2014 Chinese Ministry of Water Resource Program (201401031); the 13th Five Year Research Program from the Ministry of Science and Technology, China (SQ2016YFSF020003).

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
