# Peer review of "Monitoring surface water quality using social media in the context of citizen science"

_Hydrology and Earth System Sciences, 2016_

## Referee Comment (RC1) · Anonymous Referee #1 · 23 Sep 2016

The manuscript by Zheng et al. describes an innovative and interesting citizen science - volunteered geographic information (VIG/CS) project. The project used observational qualitative methods to describe water quality. The project has high potential for success. With additional data, quality control and analysis, the authors will make important contributions to changes in river quality, citizen science in China and the role of cash prizes in VIG/CS engagement. However, at the moment, the results presented are extremely limited, too limited to allow for any significant discussion or analysis. The validation of the method is based on 15 samples taken by experts. The comparison between different indices shows only that some relationship exists between the data and there are not real analysis of the type of the data (normally distributed, significance, ..). It is unclear how the cs based index are related to the national values as they use two different scales, a simple regression is not sufficient. The larger Mode 1 data was

not used, either for single parameter analysis (were blooms reported in the months when blooms occur on the Yellow river)? or in their aggregate index (does the trend reported reflect expected spatial trends?). In general, there are not enough validation data to demonstrate that the method is useful as a tool for river quality analysis. The possibility to show the utility of cash prizes as an engagement approach is valuable, but again, there are not enough data. I suggest that the authors wait until they have enough data or reduce this manuscript into a short communication and concentrating on one robust result.

---

## Short Comment (SC1) · 2 Oct 2016

General comments

This manuscript introduces an innovative way of monitoring surface water quality using citizen contributed observations and social media. The study falls in an emerging category of research in environmental management that focuses on combining the potentials of Information Communication Technologies (ICTs) and citizen science activities. The text is rather well-written and structured (with minor exceptions that are discussed in the 'technical corrections' section of this review). There are some references missing along the text (see specific comments). The higher objective of the research is to propose a "framework and method" that "can provide a mechanism to collect water quality data from citizens and offer a primary foundation for big data analysis in future

research", however, there are a number of methodological and data analysis ambiguities that for sure can benefit from further clarification and discussions (see specific comments in the next section). Thus, it is very much important that the authors explain the assumptions and choices made in interpreting the citizen contributed data.

Specific comments

1- There are a number of arguments along the text that tend to oversimplify numerous social and technical difficulties of citizen engagement in environmental monitoring, here are some examples:

Page 2 (line#9) authors claim that "If a sufficient number of volunteer reporters (e.g., the two citizens) come forward, then the hidden sewage dumping and pollution can be detected with considerably low cost"; It should be noted that 'coming forward' and getting engaged in environmental monitoring, does not necessarily mean that the volunteers will remain engaged in the activity in long run. Quality Control measures should also be studied, and discussed carefully before making such claims; false data may result in poor, costly, and sometimes irreversible decisions.

Page 6 (line#9) authors claim that "If these people log in and submit reports, then they become volunteers. Thereafter, they may share their reports and attract numerous volunteers to be involved"; this sentence sound like a claim by the authors, and can also benefit from a 'more careful' restatement as it (perhaps unintentionally) undermines the difficulties involved in citizen engagement.

2- On Page 6 (line#9) authors mention that "Professionals who work for environmental authorities and organizations were interviewed and convinced to register in TEMP"; is it possible to provide a brief conclusion from the findings of these interviews? For example, what were their incentives for participation? Or, how they were 'convinced'?

3- I find it difficult to understand the concept behind formulating the equation (1) on page 7. I found this equation problematic as according to the formula S, T, and F (that

are among the physical characteristics of the water) are 3 times less important than 'Ia' that is solely based on data collector's perspective. The authors should explain more about the assumptions behind formulating this equation as they highly affect the final results of this research.

4- On page 10 (line#15) authors conclude that "this difference may be attributed to that citizens tend to overrate the water quality". In line with the previous comment; how does this conclusion change if we gave the same weight to all attributes in formula (1) on page 7?

5- All of the attributes discussed in the graphs on page 11 (floats, water smell, and turbidity) are highly dependent on the time of observation, and also the location. How did the authors include the 'time of observation' and 'location of the samples' in comparing official and volunteer data?

6- On Page 19 (line#10) authors refer to the concept of feedback loop in participatory processes without introducing it earlier in the paper. This concept needs to be introduced at an earlier stage in the paper.

7- References are missing in the following sections:

Page 1 (line#28); "Currently, collecting a single sample at a site costs between US$4,000 and US$6,000"; reference is missing

Page 9 (Line#7); "Among the 83 cross-sections surveyed during the Yellow River quality assessment in 2004, 72.3% felt into the Grade III quality standard"; reference is missing

Page 9 (Line#15); "The hydrological regime at this station represents an overview of the hydrological regime of the entire river basin"; Authors' claim, without reference

Technical corrections

Page 5(Line#29); change "WeChat in their mobile devices" to "WeChat on their mobile

devices".

Page 10(Line#6); Typo; should be Figure 3.

Figure 3(a) on Page 10; add vertical axes title/unit.

Check section 4.2 on page 11 for typos and errors, here are some examples ( (1) Beijing has 29 reports not 30; (2) Table 3 shows the overview of the 172 validated reports, not the reports themselves; (3) Based on the content of Table3 this statement is not true: "A total of 10 provinces have under 10 reports"; (4) Based on the content of Table3 this statement is not true:"A total of 13 provinces have only 1 report during the period" ).

---

## Author Comment (AC1) · 4 Oct 2016

Thank you for the comments. We agree that it is not sufficient for comparison and regression using 15 samples. We reduce the manuscript and remove the section about the data comparison in the Yellow River. Deeper investigation will be conducted to analysis the relationships between CS based data and national values in future. In the current revised paper, we concentrated on analyzing the behaviors of volunteers basing on the reports over the China, including the spatial distribution of the reports, the proportion of anonymous reports, and the utility of cash prizes, etc. Referring to the cash prizes, the number of volunteers increased more 3 times after the cash prizes. It should be demonstrated as an effective approach for volunteer engagement.

We believe that the proposed method is valuable and practicable for river quality mon-

itoring, as the platform already has more than 600 users and received about 200 reports. The methods for volunteer's recruitment, engagement and encouragement are also described and discussed basing on the reports. The effectiveness of Mode 1 and Mode 2 is analyzed according to the increase of the number of the volunteers. The approaches for data washing is also provided in the Discussion section. These contents could provide ideas and references for more and further research in this area. Therefore, we think that the revised manuscript is probably more than a short communication and is still meaningful for a technical paper.

Please see the attached revised manuscript.

Please also note the supplement to this comment:
http://www.hydrol-earth-syst-sci-discuss.net/hess-2016-359/hess-2016-359-AC1-supplement.pdf

**Supplement:**

[revised manuscript text omitted]

20   In Mode 2, a group of professional citizens were recruited to monitor the water quality in targeted sites. Professionals who work for environmental authorities and organizations were interviewed and convinced to register in TEMP. They were required to monitor water bodies that they are familiar with and regularly submit the reports through TEMP. In Mode 2, volunteers were recruited and data were instantly collected before an extensive range of citizens who were involved through Mode 1.

25   ### 2.4 Data analysis

A method was established to quantitatively analyse monitoring reports. Smell, turbidity, floats, and integrated assessment reported by the volunteers were quantified and normalized between 0.0 and 1.0 according to their ranking scores. Water colour was used for data cleansing and rumour control through cross validation between the descriptions and photos. Table 1 shows the indicators and their value ranges.

30

**Table 1: Normalization of indicators**

| Report items | Indicator | Data type | Qualifications |
|---|---|---|---|
| Water color | $C$ | Text | None |
| Smell | $S$ | Score from 0 to 10 | 1.0–0.0 |
| Turbidity | $T$ | Score from 0 to 10 | 1.0–0.0 |
| Floats | $F$ | Score from 0 to 10 | 1.0–0.0 |
| Integrated assessment | $I_a$ | Grand from 1 to 6 | 1.0, 0.75, 0.5, 0.25, 0.0 |

The value of smell, turbidity, and floats ranges from 1.0 to 0.0. The integrated assessment's indicator is normalized across 1.0, 0.75, 0.5, 0.25, and 0.0, which correspond to the five grades of water quality assessment (i.e., excellent, good, bad, very bad, and worst).

$$Q = (P_a + I_a)/2 = [(S + T + F)/3 + I_a]/2, \tag{1}$$

where $Q$ represents the report's water quality assessment result. A preliminary assessment ($P_a$) is calculated through the average of $S$, $T$, and $F$. The average value of $S$, $T$ and $F$ ($P_a$) provides an indicator based assessment of the water quality. $I_a$ is an integrated assessment in the reports based on reporters' overall perception of the water. t is higher than the individual sense based indicators $S$, $T$, and $F$. 
[revised manuscript text omitted]

---

## Author Comment (AC2) · 4 Oct 2016

General comments This manuscript introduces an innovative way of monitoring surface water quality using citizen contributed observations and social media. The study falls in an emerging category of research in environmental management that focuses on combining the potentials of Information Communication Technologies (ICTs) and citizen science activities. The text is rather well-written and structured (with minor exceptions that are discussed in the 'technical corrections' section of this review). There are some references missing along the text (see specific comments). The higher objective of the research is to propose a "framework and method" that "can provide a mechanism to collect water quality data from citizens and offer a primary foundation for big data analysis in future research", however, there are a number of methodological and data analysis ambiguities that for sure can benefit from further clarification and dis-

cussions (see specific comments in the next section). Thus, it is very much important that the authors explain the assumptions and choices made in interpreting the citizen contributed data.

Reply: Thanks for your comments. We agree that there are a number of ambiguities should be explained and clarified. We reduce the manuscript and remove the section about the data comparison in the Yellow River, because there are only 15 simples in the Yellow River. Deeper investigation will be conducted to analysis the relationships between CS based data and national values in future, when we receive more data. In the current revised paper, we concentrated on analyzing the behaviors of volunteers basing on the reports over the China, including the spatial distribution of the reports, the proportion of anonymous reports, and the utility of cash prizes, etc.

Specific comments 1- There are a number of arguments along the text that tend to oversimplify numerous social and technical difficulties of citizen engagement in environmental monitoring, here are some examples:

Page 2 (line#9) authors claim that "If a sufficient number of volunteer reporters (e.g., the two citizens) come forward, then the hidden sewage dumping and pollution can be detected with considerably low cost"; It should be noted that 'coming forward' and getting engaged in environmental monitoring, does not necessarily mean that the volunteers will remain engaged in the activity in long run. Quality Control measures should also be studied, and discussed carefully before making such claims; false data may result in poor, costly, and sometimes irreversible decisions.

Reply: Yes, the participation of the volunteer does not mean the continuous engagement. We have revised the sentence to "If there are more volunteer reporters (e.g., the two citizens) come forward, the more possibility to detect the hidden sewage dumping or pollution." (Page 2 ,line#9). Sufficient number of volunteers and reports could provide the possibility and foundation for analysis, including the quality control and data mining.

Page 6 (line#9) authors claim that "If these people log in and submit reports, then they become volunteers. Thereafter, they may share their reports and attract numerous volunteers to be involved"; this sentence sound like a claim by the authors, and can also benefit from a 'more careful' restatement as it (perhaps unintentionally) undermines the difficulties involved in citizen engagement.

Reply: Yes, if the people log in the platform, they may become a volunteer. But it is still difficult to make them submit the reports continuously. We change this expression to "If these people log in and submit reports, they will be involved and contacted by the TEMP. They can receive the message from TEMP and be encouraged to summit more reports and attract numerous volunteers to be involved." (Page 6,line#16)

2- On Page 6 (line#9) authors mention that "Professionals who work for environmental authorities and organizations were interviewed and convinced to register in TEMP"; is it possible to provide a brief conclusion from the findings of these interviews? For example, what were their incentives for participation? Or, how they were 'convinced'?

Reply: These professionals involved in the TEMP at current stage are research collaborators and colleagues of the authors. We introduced the platform to them through Internet and they agreed to use the platform and were involved in our research. Their incentives are mainly the interests on environment research. Further interview will be conducted to collect the feedbacks from these professionals in future.

3- I find it difficult to understand the concept behind formulating the equation (1) on page 7. I found this equation problematic as according to the formula S, T, and F (that are among the physical characteristics of the water) are 3 times less important than 'Ia' that is solely based on data collector's perspective. The authors should explain more about the assumptions behind formulating this equation as they highly affect the final results of this research.

Reply: Thanks for your comments on this. Ia is an integrated assessment in the reports based on reporters' overall perception of the water. It is higher than the individual

sense based indicators, S, T, and F. The average value of S, T and F (Pa) provides an individual based assessment of the water quality. We believe that Ia is more integrated and more important than S, T and F. (Page 7, Line10)

4- On page 10 (line#15) authors conclude that "this difference may be attributed to that citizens tend to overrate the water quality". In line with the previous comment; how does this conclusion change if we gave the same weight to all attributes in formula (1) on page 7?

5- All of the attributes discussed in the graphs on page 11 (floats, water smell, and turbidity) are highly dependent on the time of observation, and also the location. How did the authors include the 'time of observation' and 'location of the samples' in comparing official and volunteer data?

Reply: Thanks for your comments. We reduce the manuscript and remove these content about the data comparison in the Yellow River. Deeper investigation will be conducted to analysis the relationships between CS based data and national values in future, when we receive more data. In the current revised paper, we concentrated on analyzing the behaviors of volunteers basing on the reports over the China, including the spatial distribution of the reports, the proportion of anonymous reports, and the utility of cash prizes, etc.

6- On Page 19 (line#10) authors refer to the concept of feedback loop in participatory processes without introducing it earlier in the paper. This concept needs to be introduced at an earlier stage in the paper.

Reply: We have move the feedback loop to the Introduction Section.

  7- References are missing in the following sections:

Page 1 (line#28); "Currently, collecting a single sample at a site costs between US$4,000 and US$6,000"; reference is missing.

Reply: The reference is "Horowitz, A. J.: A review of selected inorganic surface water

quality-monitoring practices: Are we really measuring what we think, and if so, are we doing it right?, Environ. Sci. Technol., 47(6), 2471–2486, doi:10.1021/es304058q, 2013." (Page 18, line 15)

Page 9 (Line#7); "Among the 83 cross-sections surveyed during the Yellow River quality assessment in 2004, 72.3% felt into the Grade III quality standard"; reference is missing Page 9 (Line#15); "The hydrological regime at this station represents an overview of the hydrological regime of the entire river basin"; Authors' claim, without reference

Reply: We have remove this section in Page 9.

Technical corrections

Page 5(Line#29); change "WeChat in their mobile devices" to "WeChat on their mobile devices".

Page 10(Line#6); Typo; should be Figure 3. Figure 3(a) on Page 10; add vertical axes title/unit.

Check section 4.2 on page 11 for typos and errors, here are some examples ( (1) Beijing has 29 reports not 30; (2) Table 3 shows the overview of the 172 validated reports, not the reports themselves; (3) Based on the content of Table3 this statement is not true: "A total of 10 provinces have under 10 reports"; (4) Based on the content of Table3 this statement is not true:"A total of 13 provinces have only 1 report during the period" ).

Reply: Sorry for the mistakes. We have revised these in the manuscript.

Please find the revised manuscript in the attachment.

Please also note the supplement to this comment:
http://www.hydrol-earth-syst-sci-discuss.net/hess-2016-359/hess-2016-359-AC2-supplement.pdf

[Figure]

**Supplement:**

[revised manuscript text omitted]

20   In Mode 2, a group of professional citizens were recruited to monitor the water quality in targeted sites. Professionals who work for environmental authorities and organizations were interviewed and convinced to register in TEMP. They were required to monitor water bodies that they are familiar with and regularly submit the reports through TEMP. In Mode 2, volunteers were recruited and data were instantly collected before an extensive range of citizens who were involved through Mode 1.

25   ### 2.4 Data analysis

A method was established to quantitatively analyse monitoring reports. Smell, turbidity, floats, and integrated assessment reported by the volunteers were quantified and normalized between 0.0 and 1.0 according to their ranking scores. Water colour was used for data cleansing and rumour control through cross validation between the descriptions and photos. Table 1 shows the indicators and their value ranges.

30

**Table 1: Normalization of indicators**

| Report items | Indicator | Data type | Qualifications |
|---|---|---|---|
| Water color | $C$ | Text | None |
| Smell | $S$ | Score from 0 to 10 | 1.0–0.0 |
| Turbidity | $T$ | Score from 0 to 10 | 1.0–0.0 |
| Floats | $F$ | Score from 0 to 10 | 1.0–0.0 |
| Integrated assessment | $I_a$ | Grand from 1 to 6 | 1.0, 0.75, 0.5, 0.25, 0.0 |

The value of smell, turbidity, and floats ranges from 1.0 to 0.0. The integrated assessment's indicator is normalized across 1.0, 0.75, 0.5, 0.25, and 0.0, which correspond to the five grades of water quality assessment (i.e., excellent, good, bad, very bad, and worst).

$$Q = (P_a + I_a)/2 = [(S + T + F)/3 + I_a]/2, \tag{1}$$

where $Q$ represents the report's water quality assessment result. A preliminary assessment ($P_a$) is calculated through the average of $S$, $T$, and $F$. The average value of $S$, $T$ and $F$ ($P_a$) provides an indicator based assessment of the water quality. $I_a$ is an integrated assessment in the reports based on reporters' overall perception of the water. t is higher than the individual sense based indicators $S$, $T$, and $F$. 
[revised manuscript text omitted]

---

## Referee Comment (RC2) · Anonymous Referee #2 · 18 Nov 2016

This is an interesting paper that addresses the question of using crow sourcing to obtain water quality reports and thereby identify pollution issues. The paper implements a platform on on We-Chat in China. It is generall well written (there are suggested wording changes in the marked up manuscript).

I found the introduction to be well written but there was little discussion of the many exisiting citizen science water quality programs. A quick scopus search revealed a number of papers with title that appear relevant. This context and the learning from those studies would need to be integrated in here.

While the ideas are sound, there is little evaluation data. The quality of reports is assessed using a small group of contributors with water quality expertise specifically recruited to the project and with experience of the site they were making reports from.

A wider group of reporters is also discussed but the quality of their reports can't be validated. I think this evaluation needs to be extended somehow, and in particular the reporting by general untrained users would need to be evaluated for this to be a publishable paper. I would encourage the authors to pursue that option as the platform and approach seems like a sound idea - but it needs better evaluation.

I have incldued a range of more detailed comments to help the authors improve their manuscript in the marked up version of the paper.

Please also note the supplement to this comment:
http://www.hydrol-earth-syst-sci-discuss.net/hess-2016-359/hess-2016-359-RC2-supplement.pdf

[Figure]

**Supplement:**

[revised manuscript text omitted]

Sensory Indicator:
- Color, description
- Smell, ranking scale
- Turbidity, ranking scale
- Floats, ranking scale
- Integrated assessment

Anonymous report?  Yes / No

Submit current positon?  No / Yes

Data collection:
- Authorization enquire
- Manually input
  - By button click → Color, smell, turbidity….
  - By typewriting → Textbox for description
  - By camera → Photograph the water

Extract user name
Extract GPS positon

Communication → Publish, retweet, share, comment on the monitoring reports

Display → Plot the spatial distribution map of the reports and pictures

Incentives → User ranking by contribution and rewards on contributors

Data download → Website for data display, management, and download

Mode 1    Mode 2

Data collect:
- Receive data from volunteers
- Collect the targeted data
- Economic incentives
- Multiple source in social media

Analysis:
- Comparison with the gauged water quality data
- Detection of sewage dumping events and pollution warning

Outcomes → A framework and a tool to use social media in monitoring water quality by citizens

[revised manuscript text omitted]

---

## Author Comment (AC3) · 28 Nov 2016

This is an interesting paper that addresses the question of using crow sourcing to obtain water quality reports and thereby identify pollution issues. The paper implements platform on We-Chat in China. It is generally well written (there are suggested wording changes in the marked up manuscript). I found the introduction to be well written but there was little discussion of the many existing citizen science water quality programs. A quick scopus search revealed a number of papers with title that appear relevant. This context and the learning from those studies would need to be integrated in here. While the ideas are sound, there is little evaluation data. The quality of reports is assessed using a small group of contributors with water quality expertise specifically recruited

to the project and with experience of the site they were making reports from. A wider group of reporters is also discussed but the quality of their reports can't be validated. I think this evaluation needs to be extended somehow, and in particular the reporting by general untrained users would need to be evaluated for this to be a publishable paper. I would encourage the authors to pursue that option as the platform and approach seems like a sound idea - but it needs better evaluation. I have included a range of more detailed comments to help the authors improve their manuscript in the marked up version of the paper. Please also note the supplement to this comment: http://www.hydrol-earth-syst-sci-discuss.net/hess-2016-359/hess-2016-359-RC2- supplement.pdf

Reply: Thank you for your comments. We revised the manuscript accordingly. 1. A literature review on community-based water quality monitoring programs are presented in the Introduction section of revised manuscript.

2. We fully agree that the data needs to be evaluated somehow. First, it is not sufficient for comparison and regression using 15 samples to evaluate the monitoring data in the Yellow River. We reduced the manuscript and removed the section of data comparison in the Yellow River. In the current revised paper, we concentrate on analyzing the behaviors of volunteers basing on the reports over the China, including the spatial distribution of the reports, the proportion of anonymous reports, and the utility of cash prizes, etc.

Second, we evaluate the monitoring data through the photos of water bodies. The monitoring reports with photos are regarded as higher credible data than those without photos. It is very difficult to validate the volunteers' monitoring data across China, because the volunteers distribute all over the country and they submitted the reports randomly once they saw the dirty water nearby. There is rarely an official gauge site just locating on a volunteer's report point. The current manuscript demonstrate that the proposed framework and platform are practicable to collect water quality monitoring data through social medial. It provides a foundation to validate the data through big data method if we can obtain a huge amount of reports in future.
3. The details comments are responded as follows. (1) The reviewer suggested that "Rather than giving the results here, perhaps rephrase to say what issues were analyzed." in Page 4, Line 25~27. We have deleted the sentences "Results revealed a spatiotemporal relationship between social media messages and real-world environmental status changes. The results also suggested new methods to monitor water pollution using VGI and social media in the future." We added "The impacts of photographed function, anonymous submission and economic incentives on increasing data credibility and volunteer motivation are analyzed".

(2) The reviewer mentioned that "This rating scale seems to be the reverse of the others where poor quality was 10" at Page 5, Line 15. Number 10 means the water is extremely smelly, completely turbid and full covered by floats. We believe it is convenient for volunteers using larger number to express heavier pollutants in and on the water.

(3) The reviewer asked that "Not clear - does this mean Mode 2 was done before Mode 1? If so why not reverse the order?" in Page 6, Line 15. We have revised this expression to "In Mode 2, The TEMP can recruit specific volunteers and collect a number of data more quickly than in Mode 1."

(4) The reviewer asked that "Were the observers aware of the official water quality test results from the days before each Q observation? This is important as it could impact on their perceptions." " in Page 10, Line 7. In this section, we agree that the data is limited to verify the volunteers' reports. We remove the section in the revised manuscript.

(5) The reviewer suggested that "Remind the reader about the time frame - start and end of the reporting included here." in Page 11, Line 9. We changed the text to "TEMP received a total of 265 reports from volunteers in China between 12 October 2015 and 30 May 2016. Out of the 265 reports, 172 were validated in the present study."

(6) The reviewer asked that "Is it 10 provinces with between 2 and 9 reports?? If 13

have only 1, then there must be at least 13 with less than 10!" in Page 11, Line 10. Sorry, there is a mistake. We have revised the text to "A total of 23 provinces have under 10 reports. A total of 14 provinces have only 1 report during the period."

(7) The reviewer asked that "Is there any means of comparing some of the reports in this section to WQ data from stations?" in Page 13 Table 2. It is quite hard to compare the reports to the gauged data across China. The volunteers submit the reports when they meet the dirty water. It could be anywhere and anytime. The coverage of gauged sites is not enough for comparison. If we have plenty of volunteer reports, it is possible to select some reports and compare them with the nearest stations' gauged data.

(8) The reviewer asked and suggested that "In the longer term if this is widely successful, how would photo interpretation be automated? This is an issue for discussion section." in Page 13, Line 9. We have added a paragraph in the Discuss section to discuss the potential of automatic photo interpretation and smartphone-based external devices for water quality monitoring.

(9) The reviewer required to explain what red packet money is in Page 17. We added more details about the red packet money on Line 30 Page 14 in the revised manuscript.

(10) The reviewer suggested that some sentences in the Results Section could be better in Discussion, including "A few volunteers were concerned about their privacy; thus, they submitted 15 reports anonymously. The number of anonymous reports implies the necessity of setting up an anonymous function in TEMP because volunteers care about their privacy when they disclose the water pollution activities around them. (Page 11, Line 15)", "The volunteers were also concerned about the charge of the Internet data flow, through which they upload the photos without Wi-Fi. Additional incentive measures are required to encourage volunteers to upload the photos of water (Page 12 Line 3~4)". We believe that those sentences provide explanations for the results and it is close to the Results section.

(11) The reviewer suggested to move the Table 3 in Page 13 to the supplementary

material. Table 3 is the main result of this study and we think it is better to keep it in the main body of the manuscript.

(12) The manuscript were edited according to the reviewer's comments on wording.

Please also note the supplement to this comment:
http://www.hydrol-earth-syst-sci-discuss.net/hess-2016-359/hess-2016-359-AC3-supplement.pdf

[Figure]

**Supplement:**

[revised manuscript text omitted]

There is an opportunity to develop image analysis to check if the photo matches the ratings of turbidity. E. Minkman; P.J. van Overloop; M.C.A. van der Sanden, 2015 explored the potential of citizen science in mobile crowd sensing in water quality monitoring, using mobile crowd sensing application for water quality measurements in the Netherlands. It consists of a colorimetric analysis using smartphone camera. The indicator test strips are photographed and the colour of the strips are analysed automaticity by an App in the smartphone to obtain chemical water quality data. Snik et al., 2014  developed the iSPEX, a low-cost, mass-producible optical add-on for smartphones with a corresponding app. People could purchase the

iSPEX smartphone accessory and place it on top of their iPhone. They are able to measure particulate matter in the atmosphere using a special smartphone application (Minkman, 2015). Such devices are meaningful in citizen-based water quality monitoring to increase data credibility. The devices should be smart, portable and convenient to use with smartphone, 
[revised manuscript text omitted]

---

## Author Response (AR1)

Editor Decision: Reconsider after major revisions (further review by Editor and Referees) (14 Dec 2016) by Pieter van der Zaag

Comments to the Author:

I find the topic of the paper very interesting and potentially very valuable. However, the paper needs a major revision because of the following four comments, whereby my Comment 1 is the most important.

**Comment 1**

I agree with the two reviewers that on the one hand, the validation of the methodology in the original manuscript lacked robustness, but that on the other hand, without such a validation the manuscript does not have sufficient content to be publishable in HESS.

I therefore encourage the authors to see whether there are any gauged water quality data that can be linked to either one of the three parameters that have been reported by citizens , i.e. either C (water colour), or S (smell) or T (turbidity). If so, then the gauged data could be compared with the citizen contributed data.

I wasn't convinced by the method adopted namely to compare the three citizen reported parameters (S, T, F) with the combined indicator Q that collapses the scores for E, N and D through averaging.

I would think that turbidity would often be routinely measured; if so that could then be a way to validate, albeit on only one parameter, the VGI method discussed in the paper.

So just throwing out the validation section altogether is not the way forward to get the paper published in HESS.

**Comment 2**

I do not understand the usefulness of equation 1, which combines all information on different parameters into one aggregate indicator Q through averaging. The question is: does this create additional information or does this in fact result in a loss of information. Also, why average the scores for C, S, and T? One could also aggregate in a different way, such as Min (S,T,F) or Max (S,T,F). And why combine the, arguably more "subjective", indicator for integrated assessment with that of the three parameters? And if we have to combine them, again why take the average?

I would find it much more interesting to compare the scores for S with Ia, F with Ia and F with Ia, and see whether there are differences in correlations, and whether there can be explanations for these differences (or similarities).

**Comment 3**

In my opinion Figure 9 adds little information, and can in my view be omitted.

**Comment 4**

There is need for strict language editing by a native English speaker.

I hope the authors consider my decision in a constructive light. I encourage them to find a way to validate the claim that citizens can indeed contribute data that are valuable in terms of water quality monitoring.

**Responses to the comments:**

Thanks a lot for the comments. We truly agree that it is important to increase the robustness of validation. We have provided more data and analysis for method validation.

**To comment 1:**
We tried to improve the validation by providing more data, comparing the reported water quality between upstream and downstream regions, analyzing the correlations between *S*, *T*, *F* and *Ia*, and comparing the reported turbidity with the gauged data.

(1) The collected data has been extended to 15 December 2016. The TEMP is continuously receiving the water quality reports during 2016. Comparing with the former manuscript, more data is provided for validation. 219 reports are analyzed after data cleansing.

(2) The Figure 4 in the revised manuscript presents the average value of *Ia* in each province of China. The reported water quality in the provinces located upstream Yellow River, Yangzi River, and Pearl River, such as Tibet, Qinghai, and Yunnan, is better than those observed at downstream provinces, such as Shandong and Guangzhou. This fits the general situation of water quality along a river. This finding also illustrates the water quality situation in China and the reasonableness of the VGI reports.

(3) The correlations between *S*, *T*, *F* and *Ia* are presented in the revised manuscript. Figures 5, 6, and 7 analyze the smell (*S*), turbidity (*T*), floats (*F*), and integrated assessment (*Ia*) of the water bodies presented in 107 reports with photographs. These reports are divided to five groups according to *Ia*, namely, 0.00, 0.25, 0.50, 0.75, and 1.00. The reports in the same group have the same value of *Ia*. Figures 5, 6, and 7 plot the minimum, maximum, and average *T*, *F*, and *S* for each group, respectively. *Ia* is highly correlated with *T*, which suggests that the VGI reports are reasonable. The volunteers have completed the reports based on their actual observations of the water. *T* has a higher correlation with *Ia* than *F*. Water turbidity can be easily observed and greatly influence the judgment of volunteers compared with the other indicators. People tend to rate the quality of muddy water as bad despite the absence of floating objects.

(4) The reported turbidity is compared with the gauged data at Huayuankou station on the Yellow River. The reported and gauged data show similar temporal variations of *T*, which indicates the effectiveness of the VGI data to some extent.

TEMP recruits a group of professional volunteers from YRCC who continuously report the quality of water at the Huayuankou station where gauged data are available. These volunteers are familiar with the quality of water on the site, but are not oriented to the gauged data during their submission of reports. The gauged results can be assessed at least one day after sampling at the station because the water sample must be analyzed beforehand in a laboratory. The volunteers can only assess the gauged data after submitting their reports. TEMP records the time of the reports according to the clock on the TEMP server. The time of the report cannot be modified by the volunteer. TEMP has only received 13 reliable photograph reports at the station between March and April 2016. Despite the limited data, the validation indicates that the VGI data are valuable for water

quality monitoring to some extent.

**To comment 2:**

Thanks again for the comment. We totally agree that it can provide more information to compare the scores of *S* with *Ia*, *T* with *Ia* and *F* with *Ia*. We analyze the correlations between Min (*S*), Average(*S*), Max (*S*) and *Ia*; Min (*T*), Average (*T*), Max (*T*) and *Ia*; Min (*F*), Average (*F*), Max (*F*) and *Ia*. See Figures 5, 6, and 7 in the revised manuscript. The results show that the *F* is highly correlated with *Ia.* This suggests that the turbidity is the key element effecting people's sense on water quality. Water turbidity can be easily observed and greatly influence the judgment of volunteers compared with the other indicators. This also demonstrates that the VGI reports are reasonable, because the volunteers indeed completed the reports based on their actual observations of the water.

**To Comment 3**

We have removed the Figure 9 from the former manuscript. .

**To Comment 4**

The revised manuscripts is edited by a native English speaker.

The department of author Hang Zheng, has change to:

[revised manuscript text omitted]
 (BVHP) community in San Francisco, California. These programs provide the protocols, guidelines, equipment and trainings to volunteers for water quality monitoring. The volunteers sample the water and measure the quality indicators through test strips and apparatus. The economic cost and inconvenience of community-based monitoring are still high and it limits the application of the program.

10   Social media, such as Twitter, Facebook, Sina Weibo, and Tencent Weixin (the popular Chinese version of Twitter), is capable of guiding and providing incentives to volunteers through real-time online communication among volunteers. In recent years, social media has become a major communication channel in our society (Jiang et al., 2015). Social media refers to Internet-based applications that enable people to conduct online communications intended for interaction, community input, and collaboration (Lindsay, 2011). Social media enables information sharing from multiple parties on computers and

15   mobile devices, particularly through social networking sites (e.g., Facebook, YouTube, and Twitter), texting, chat rooms, discussion forums, and blogs (Tang et al., 2015). Social media builds on the ideological and technological foundations of Web 2.0, and enables the creation and exchange of user-generated content (Kaplan and Haenlein, 2010). The major functions of social media in the environmental management processes include one-way information sharing, two-way information sharing, situational awareness, rumor control, reconnection, and decision-making (Tang et al., 2015). Jiang et al. (2015)

20   effectively monitored the dynamic changes of air quality in large cities by analysing the spatiotemporal trends in geo-targeted social media messages with comprehensive big data filtering procedures. Werts et al. (2012) established a website called AbandonedDevelopments.com to collect VGI and detect the sediment pollution of abandoned structures in upstate South Carolina. It combined Web-GIS technologies, data sources, and social media for future use in soil and water conservation.

The advertising, instruction, and guidance for water quality monitoring can be spread extensively and delivered to the potential volunteers' mobile devices directly through social media platforms. The observed sewage dumping or water pollution activities can be disseminated rapidly in social media networks and call the government's attention. Social media provides the platform for volunteers to present, discuss, and communicate their criticism, anger, and solutions to the water

30   pollution they observe. Communication and mutual encouragement provide a strong motivation for volunteers to monitor water quality and persistently share their observations. Furthermore, the discussion on the pollution activities reported in social media networks increases the opinion's pressure on the government to solve the problem. Government feedback can also be promptly disseminated to volunteers through social media. Timely dissemination of government feedbacks also motivates volunteers to continuously monitor water quality (see Fig. 1). Today, volunteers are equipped with digital cameras,

GPS, digital maps, and other resources through their smartphone. Multiplying the resources of the average empowered citizen by the population of the city results in an astounding ability to create and share information (Goodchild and Glennon, 2010).

[Figure]

5  Figure 1: Positive feedback ring for volunteer's motivation.

This study aims to establish an approach to monitor surface water quality through volunteered citizen scientists. An App platform basing on social media is built to collect the water quality information. The application of the App suggests new methods to monitor water pollution using VGI and social media in the future. Results reveal the feasibility of VGI in monitoring water quality. The impacts of photographed function, anonymous submission and economic incentives on
10  increasing data credibility and volunteer motivation ares analysed. This paper is organized as follow. Methodology is presented in Section 2. Monitoring reports obtained across China are displayed in Section 3. The data quality and motivation of volunteers are discussed in Section 4. Conclusions drawn from this study are given in Section 5.

**2 Methodology**

A methodological framework is established to: (1) Collect sensory data of surface water quality from the volunteer citizens,
15  who describe and photograph the water that they pass by or are close to. Descriptions and photos are sent to a data centre from their mobile devices of volunteers through a specific social media application in the social media platform. (2) DetectDisclose the illegal/hidden sewage dumping that is very difficult to detect through the official monitoring network. Once sewage dumping reports from the volunteers are submitted, the data centre will transfer them to the mass media and water authorities after undertaking credibility checks.

**2.1 Data type**

Four indicators  are adopted to describe the physical characteristics of water quality (see Figure 1). A total of 11 water color options  are provided for volunteers to choose from, including red, orange, yellow, green, cyan, blue, purple, milky, pink, black, and crystal. The second indicator is smell, which is quantified by the ranking scores made by volunteers.

5   The volunteer is asked to rank the smell of the sample from 0 to 10, with 0 implying a lack of odour and 10  representing a foul-smelling sample. The third indicator is turbidity in which a score of 10 means that the water is non-transparent and a score of 0 represents transparency. The higher score claims that there are more contaminants in the water. The fourth indicator represents floating objects or floating material on the water. If the water is completely covered by oil, plastics, and rubbish, among others, the floats score is 10. By contrast, a score of 0 means no floating object is present. The

10   last item is an integrated assessment on the water quality, ranking from worst, very bad, bad, and good, to excellent. Volunteers evaluate the water quality based on their perception.

**2.2 App in social media platform**

The Tsinghua Environment Monitoring Platform (TEMP, http://www.thuhjjc.com/) application  is built based on WeChat public accounts. WeChat is a mobile text and voice messaging communication service developed by Tencent in

15   China, and was first released in January 2011. WeChat is one of the largest messaging applications in China. As of May 2016, WeChat has over a billion existing accounts and 700 million active users (Intelligence, 2016). WeChat provides text messaging, hold-to-talk voice messaging, broadcast messaging, video conferencing, photo and video sharing, and location sharing (Tencent, 2016). Moreover, WeChat enables users to register a public account, thereby enabling them to push feeds to subscribers, interact with subscribers, and provide them with services. Public accounts can be used as a service platform,

20   such as hospital pre-registrations, visa renewal, or credit card service (Wikipedia, 2016). WeChat provides functions for users to post images and texts, share music and articles, as well as comment and "like" in the Moments. Only the users' friends will be able to view the contents and comments of their Moments. In addition, WeChat supports payment and money transfer, thereby enabling users to use peer-to-peer transfer and electronic bill payment (Tencent, 2016).

25   Volunteer reporters  log into TEMP through their WeChat accounts. Through TEMP, volunteers are able to summit reports together with the GPS position where the reported water is located. The location can either be automatically extracted from the devices or be input manually by the reporters. Volunteers can tweet the reports to their friends, and post and comment on them in the Moments. TEMP also provides a function for ranking volunteers according to their contributions. Awards, such as cash through WeChat Payment,  can be offered to top-

30   ranking reporters. A computer-based website is also provided for the public to view and download the reports (See TEMP, http://www.thuhjjc.com.)

**2.3 Volunteer recruitment**

Two modes  are used for volunteer recruitment. In Mode 1, TEMP is popularized from a central group to the general public (see Figure 1). The university students recruited for the present study post the address of the TEMP two-dimension code in Moments and Chat Groups after they log in the TEMP through their WeChat accounts. Their friends who are

5 interested in SWQM will click the address or scan the two-dimension code (QR code), and then be directed to TEMP. If these people log in and submit reports, they will be involved and contacted by the TEMP.  TEMP is not able to control when and where the monitoring reports come from. Data  is scattered in Mode 1.

In Mode 2, a group of professional citizens  are recruited to monitor the water quality in targeted sites. Professionals

10 who work for environmental authorities and organizations  are interviewed and convinced to register in TEMP. They  are required to monitor water bodies that they are familiar with and regularly submit the reports through TEMP. In Mode 2, The TEMP can recruit specific volunteers and collect a number of data more quickly than in Mode 1.

**2.4 Data analysis**

15 A method  is established to quantitatively analyse monitoring reports. Smell, turbidity, floats, and integrated assessment reported by the volunteers  are quantified and normalized between 0.0 and 1.0 according to their ranking scores. Water colour  is used for data cleansing and rumour control through cross validation between the descriptions and photos. Table 1 shows the indicators and their value ranges.

**Table 1: Normalization of indicators**

| Report items | Indicator | Data type | Qualifications |
|---|---|---|---|
| Water color | $C$ | Text | None |
| Smell | $S$ | Score from 0 to 10 | 1.0–0.0 |
| Turbidity | $T$ | Score from 0 to 10 | 1.0–0.0 |
| Floats | $F$ | Score from 0 to 10 | 1.0–0.0 |
| Integrated assessment | $I_a$ | Grand from 1 to 6 | 1.0, 0.75, 0.5, 0.25, 0.0 |

Note: The value of smell, turbidity, and floats ranges from 1.0 to 0.0. The integrated assessment's indicator is normalized across 1.0, 0.75, 0.5, 0.25, and 0.0, which correspond to the five grades of water quality assessment (i.e., excellent, good, bad, very bad, and worst).

$$Q = (P_a + I_a)/2 = [(S + T + F)/3 + I_a]/2, \tag{1}$$

~~A preliminary assessment ($P_a$) is calculated through the average of $S$, $T$, and $F$. The average value of $S$, $T$ and $F$ ($P_a$) provides an indicator based assessment of the water quality. $I_a$ is an integrated assessment in the reports based on reporters' overall perception of the water. $t$ is higher than the individual sense based indicators $S$, $T$, and $F$. The water sample's smell, turbidity, and floats were used in the calculation to enhance the assessment's credibility.~~

**2.5 Validation**

The VGI data is validated through comparing the citizen-based reports with the gauged data. The reported turbidity data at Huayuankou station on the Yellow River of China is compared with the gauged data from the Yellow River Conservation Commission (YRCC). The Huayuankou station is one of the key stations on the main reach of Yellow River and it is located where the middle reach and lower reach are divided. The hydrological regime at this station represents an overview of the hydrological regime of the entire river basin.

It should be pointed out that it is quite difficult to validate all the volunteers' reports. Volunteers distribute all over the country and they submit the reports randomly once they see the dirty water nearby. There is rarely an official gauge site just locating on a volunteer's report point. The TEMP employed a group of trained volunteers to report the water quality at Huayuankou station each 2 or 3 days during March and April 2016. This provides the site-specific VGI data for validation.

[Figure]

Figure 2: Framework of monitoring water quality by VGI and social media.

**3 Results**

**3.1 Water quality reports across China**

TEMP received a total of  324 reports from volunteers in China between 12 October 2015 and  15  December 2016. Out of  those reports,  219  are adopted after data cleansing .

5    Table 2 shows the overview of 219  reports across  30 provinces and municipalities in China.  Beijing and Henan  have about 30 reports. The reports from Beijing  are mainly from the students of Tsinghua University, where the research group of the present study is located. The reports from Henan Province  are mostly contributed by Henan-based professional volunteers working in the  YRCC. There are

10  6 provinces where the number of reports is more than 10.  No reports are  provided from  Xinjiang, Hainan, Taiwan, Macau, and the South China Sea.

**Table 2: Number of reports across China**

| No. | Provinces | Number of reports | | | Integrated assessment(*Ia*)  | No. | Provinces | Number of reports | | | Integrated assessment(*Ia*)  |
|---|---|---|---|---|---|---|---|---|---|---|---|
| | | Total | Anonymous | Photographed | | | | Total | Anonymous | Photographed | |
| 1 | Beijing | 41 | 26 | 9 | 0.53 | 18 | Hebei | 3 | 1 | 0 | 0.75 |
| 2 | Henan | 37 | 19 | 17 | 0.74 | 19 | Guangdong | 3 | 3 | 2 | 0.50 |
| 3 | Shanghai | 23 | 2 | 10 | 0.71 | 20 | Qinghai | 2 | 1 | 1 | 0.88 |
| 4 | Shandong | 17 | 1 | 17 | 0.49 | 21 | Sichuan | 2 | 2 | 0 | 0.75 |
| 5 | Chongqing | 12 | 6 | 11 | 0.52 | 22 | Neimeng | 2 | 1 | 0 | 0.38 |
| 6 | Jiangxi | 12 | 3 | 4 | 0.71 | 23 | Heilongjiang | 1 | 1 | 0 | 0.50 |
| 7 | Hubei | 8 | 3 | 3 | 0.63 | 24 | Xianggang | 1 | 1 | 1 | 1.00 |
| 8 | Fujian | 8 | 3 | 3 | 0.78 | 25 | Shannxi | 1 | 1 | 0 | 0.75 |
| 9 | Gansu | 7 | 2 | 6 | 0.75 | 26 | Ningxia | 1 | 1 | 0 | 0.75 |
| 10 | Tibet | 6 | 0 | 4 | 0.83 | 27 | Jilin | 1 | 1 | 0 | 0.75 |
| 11 | Yunnan | 5 | 1 | 4 | 0.85 | 28 | Hunan | 1 | 1 | 1 | 0.50 |
| 12 | Shanxi | 5 | 3 | 3 | 0.55 | 29 | Guangxi | 1 | 1 | 1 | 0.75 |
| 13 | Jiangsu | 4 | 1 | 3 | 0.88 | 30 | Anhui | 1 | 1 | 0 | 0.75 |
| 14 | Zhejiang | 4 | 1 | 3 | 0.63 | 31 | Xinjiang | / | / | / | |
| 15 | Guizhou | 4 | 1 | 0 | 0.88 | 32 | Hainan | / | / | / | |
| 16 | Tianjin | 3 | 3 | 2 | 0.75 | 33 | Taiwan | / | / | / | |
| 17 | Liaoning | 3 | 1 | 2 | 0.58 | 34 | Macau | / | / | / | |
| Total | | | | | | | | 219 | 92 | 107 | |

Table 2 shows that 92  reports  are submitted without the WeChat users' names indicated. This represents that there are indeed a number of volunteers preferring anonymous reports due to the privacy. It is necessary  seting up an anonymous function in TEMP because volunteers care about their privacy when they disclose the water pollution activities around them. A total of 107 reports   have photos of water. Photos significantly increase the credibility of the reports by providing substantial information for water quality analysis. However, 50% of the reports  have no photos. Volunteers  are also concerned about charges for mobile Internet data usage, through which they upload the photos without Wi-Fi. Additional incentive measures are required to encourage volunteers to upload the photos of water. Figure 2 and Figure 3 show the number of reports and anonymous reports and photograph reports, respectively.

[Figure]

**Figure 2. Distribution of the reports across the provinces and cities in China.**

[Figure]

**Figure 4. Distribution of anonymous reports and photograph reports.**

[Figure]

**Figure 4. The integrated assessment ($I_a$) results of China**

The average value of $I_a$ in each province is represented in Table 2 and Figure 3. The reported water quality in the provinces locating at the source of Yellow River, Yangzi River and Pearl River, such as Tibet, Qinghai and Yunnan is better than those at downstream, e.g. Shandong and Guangzhou. This is consistent with the water quality situation in China and demonstrates the Reasonableness of the VGI reports. However, the assessment

is unable to accurately represent the overall situation of the surface water quality in a region because the coverage and frequency of the reports are insufficient. It is possible to depict . ~~The assessment results for each province were calculated based on the volunteered reports from that province (see. Equation (1)). The assessment is, however, unable to represent the overall situation of the surface water quality in a region because the coverage and frequency of the reports is insufficient at the current stage. Meanwhile, TEMP provides a practical tool for citizens to monitor the rivers and the lakes around them. An increasing number of volunteers will be involved in the future.Tor a river basin canbe depicted onceand photos~~ are provided.

**3.2 Examples of reports for pollution disclosure**

Table 3 shows three photograph reports. Algal blooms and water surface foam are shown in the photos. In Report 1, the volunteer submitted a photo of the river in Tsinghua University, Beijing, China. The river  was polluted by the domestic sewage and  suffered from eutrophication. Report 2 represents the water quality in an unidentified river located in Fei Town, Linyi City, Shandong Province. Photo 2 shows that the water surface  was covered by algae and rubbish. The reporter assessed the water quality as very bad. Report 3 is from a coastal area, namely, Tianjing City. The reporter identified the color of the water as black and assessed its quality as bad.

Among the  107 photos collected by TEMP during the period, the volunteers failed to monitor the scene, wherein the dumped sewage water  is just flowing to the river or lake. Sewage water dumping in China generally occurs at night and in a hidden place, which is rarely found by volunteers. If there are sufficient volunteers aiming to disclose the hidden sewage dumping in a region,  the pollution activities can be determined by TEMP .

**Table 3: Examples of reports for pollution disclosure**

| No. | 1 | | | | | 2 | | | | | 3 | | | | |
|---|---|---|---|---|---|---|---|---|---|---|---|---|---|---|---|
| | Color | Smell | Turbidity | Floats | Assess. $I_d$ | Color | Smell | Turbidity | Floats | $I_d$ Assess. | Color | Smell | Turbidity | Floats | $I_d$ Assess. |
| **Report** | Green | 3.0 | 5.0 | 7.0 | VB | Green | 3.0 | 6.0 | 5.0 | VB | Black | 0.0 | 7.0 | 7.0 | B |
| **Photo** |  | | | | |  | | | | |  | | | | |
| **Date** | April 10, 2016 | | | | | April 15, 2016 | | | | | May 1, 2016 | | | | |
| **Location** | Tsinghua University, Beijing City, China | | | | | Fei Town, Linyi City, Shandong Province | | | | | New coastal area, Tianjin City, China | | | | |

Note: B means bad; VB means very bad.

**3.3 Correlation between $Ia$ and $S$, $T$, $F$**

The smell ($S$), turbidity ($T$), floats ($F$) and integrated assessment ($I_a$) from 107 photograph reports are analysed and shown in Figure 5, 6 and 7. The reports are divided to five groups according to $I_a$, including 0.00, 0.25, 0.50, 0.75 and 1.00. The value of $I_a$ is equal among the reports in the same group. For each group, the minimum, maximum and average value of turbidity ($T$) is plotted in Figure 5. So does the floats ($F$) in Figure 6 and the smell ($S$) in Figure 7.

$I_a$ is highly correlated with $T$ and $S$. The correlation demonstrates that the VGI reports are reasonable. The volunteers indeed complete the reports basing on actual observation on the water. The turbidity ($T$) is higher correlated with $I_a$ than floats ($F$). Turbidity is easier to be observed and is more influencing on volunteers' judgement. People trend to regard the water as in good quality if it is crystal, despite there may be objects, e.g. leaves or brunches floating on it.

[Figure]

Table 3 s

[Figure]

Figure 5. The correlation between $I_a$ and $T$                      Figure 6. The correlation between $I_a$ and $F$

[Figure]

Figure 7. The correlation between $I_a$ and $S$

**3.4 Validation with the gauged data**

The turbidity ($T$) from the reports at Huayuankou station of the Yellow River is compared with the gauged data (see Figure 8). The temporal variation of $T$ shows to be similar between the reported and the gauged data. It demonstrates the effectiveness of the VGI data to some extent. All these reports have photos and can be regarded as high credibility. The
5  results imply that the water quality assessment through the citizens' sensory is able to be compatible with the reality.
of water

[Figure]

Figure 8. The turbidity from the reports and the gauged data

10  **5 Discussion**

The present study aimsed to develop a method to monitor surface water quality through volunteer citizens using a social media application. A framework was is established to guide the application design, volunteer recruitment, data collection, and report analysis. The TEMP application was is built based on the social media platform called WeChat. Using the application, TEMP users can describe and photograph the water in rivers and lakes following the TEMP instructions.
15  Moreover, users can report the surface water pollution activities that affect their living and health.

A total of 219 validated reports were are analyzedanalysed in this study. These reports are from 140 volunteers across 30 provinces and cities in China. The water quality on sites was is assessed by the volunteers using their sense organs, particularly through their observations on water smell, turbidity, and floating matter on the water. People's perception on
20  water quality varies, and different people may provide different assessment reports on water from the same site. Comparing the assessment results across different sites is difficult because the citizens' reports are subjective to some degree. This situation will change if numerous volunteers and extensive reports are obtained. Data could also potentially be supplemented

by satellite and aerial remote sensing and sensor system streaming. Thereafter, the big data method (Hampton et al., 2013) can be applied to improve the accuracy of water quality monitoring if high data density is present. The present study provides an approach for collecting citizen reports on water quality, which is the first step in applying the big data method in environmental governance (Perez et al. 2015; McCall 2003).

The credibility of the reports is the major concern of the present study. Regardless of the water quality assessment by citizens, identifying whether the reports are real and whether the volunteers generated the reports based on their observations is necessary rather than on false statements or rumours. Rumour control is significant when a water pollution activity is detected and reported in social media (Tang et al., 2015). The current study applied three criteria to validate wash the

10 reportsdata, including: (1) a report with the exact GPS location information of the site is regarded as credible. The sites' GPS information is automatically abstracted from the reporters' mobile devices when they submit the reports. (2) If reports are submitted several times during a short period, and most of these reports are from the same volunteer on the same site, then the reports are deemed to haveas low credibility. These reports may be test reports from new volunteers. (3) Reports with photos are the most credible. A total of 265 324 reports are cleansed following these criteria, and 172 219 were validated in

15 the present study. 107 photograph reports were regarded as high credibility. Further cross-validation among different reports can be applied if large amounts of data in a region are present.

The validation of the reports is a tricky issue. The reports are submitted from scattered sites, where there are insufficient gauged data for validation. The TEMP recruited a group of professional volunteers from the Yellow River Conservation

20 Commission. They report the water quality continuously at the Huayuankou station where the gauged data is available. They are familiar with the water quality status on site but did not know the gauged data when submitting the reports. The gauged results are assessable at least one day after sampling at the station, because the water sample needs to be analysed in the laboratory. The volunteers can only assess to the gauged data after submitting the reports. Meanwhile, The TEMP records the time of the reports according to the clock on TEMP server. There is no way for the volunteer to modify the time of the

25 report. The TEMP only received 13 reliable photograph reports at the station during March and April 2016. Although the data is limited, the validation represents that the VGI data is valuable for water quality monitoring to some extent. The citizens' assessment is effective in representing the water quality status if the reporter/citizen is relatively trained. Similar evidence was also obtained by Fore et al. (2001), Monk et al. (2008), Flanagin and Metzger (2008), and Koss et al. (2005).

30 There is an opportunity to develop image analysis to check if the photo matches the ratings of turbidity. E. Minkman; P.J. van Overloop; M.C.A. van der Sanden, 2015 explored the potential of citizen science in mobile crowd sensing in water quality monitoring, using mobile crowd sensing application for water quality measurements in the Netherlands. It consists of a colorimetric analysis using smartphone camera. The indicator test strips are photographed and the colour of the strips are analysed automaticity by an App in the smartphone to obtain chemical water quality data. Snik et al., 2014 developed the

iSPEX, a low-cost, mass-producible optical add-on for smartphones with a corresponding app. People could purchase the iSPEX smartphone accessory and place it on top of their iPhone. They are able to measure particulate matter in the atmosphere using a special smartphone application (Minkman, 2015). Such devices are meaningful in citizen-based water quality monitoring to increase data credibility. The devices should be smart, portable and convenient to use with smartphone, e.g. an external device using laser to detect water quality (Chen et al., 2015).

[revised manuscript text omitted]

A total of  324 reports across  30 provinces and cities in China were received by TEMP between October 12, 2015 and  December 15, 2016. Out of the  324 reports,  219  are validated and analyzed after data cleansing. The distribution analysis of reports across China indicates that the anonymous and photograph functions are quite essential for TEMP. Over 48% of the  219 reports are from anonymous users. Accordingly, people care about their

5 privacy when they try to disclose a water pollution activity occurring within their vicinity. A total of  107 photos of rivers and lakes  are collected through TEMP, and these photos provide extensive information for pollution detection. 13 photograph reports at the Huayuankou station in the Yellow River is validated through comparing the reported turbidity with the gauged value. It shows the citizen-based water quality data is relatively credible, if the volunteers are trained.

[revised manuscript text omitted]

---

## Author Response (AR2)

**Author's Response**

The manuscript has been revised point-by-point to the editors' comments. The changes of the manuscript can be found in the following marked-up version.

[revised manuscript text omitted]